# Emergence of alternative stable states in microbial communities undergoing horizontal gene transfer

**Juken Hong, Wenzhi Xue, Teng Wang***

Key Laboratory of Quantitative Synthetic Biology, Shenzhen Institute of Synthetic Biology, Shenzhen Institutes of Advanced Technology, Chinese Academy of Sciences, Shenzhen, China

## eLife Assessment

This manuscript offers **valuable** theoretical predictions on how horizontal gene transfer (HGT) can lead to alternative stable states in microbial communities. Using a modeling framework, **solid** theoretical evidence is provided to support the claimed role of HGT. However, given that the model has many degrees of freedom, a more comprehensive analysis of the role of different parameters could strengthen the study. Additionally, potential interactions between plasmids that carry out HGT are not discussed in the model. This paper would be of interest to researchers in microbiology, ecology, and evolutionary biology.

***For correspondence:**
t.wang1@siat.ac.cn

**Competing interest:** The authors declare that no competing interests exist.

## Abstract
Microbial communities living in the same environment often display alternative stable states, each characterized by a unique composition of species. Understanding the origin and determinants of microbiome multistability has broad implications in environments, human health, and microbiome engineering. However, despite its conceptual importance, how multistability emerges in complex communities remains largely unknown. Here, we focused on the role of horizontal gene transfer (HGT), one important aspect mostly overlooked in previous studies, on the stability landscape of microbial populations. Combining mathematical modeling and numerical simulations, we demonstrate that, when mobile genetic elements (MGEs) only affect bacterial growth rates, increasing HGT rate in general promotes multistability of complex microbiota. We further extend our analysis to scenarios where HGT changes interspecies interactions, microbial communities are subjected to strong environmental selections and microbes live in metacommunities consisting of multiple local habitats. We also discuss the role of different mechanisms, including interspecies interaction strength, the growth rate effects of MGEs, MGE epistasis and microbial death rates in shaping the multistability of microbial communities undergoing HGT. These results reveal how different dynamic processes collectively shape community multistability and diversity. Our results provide key insights for the predictive control and engineering of complex microbiota.

## Introduction

Microbial communities living in similar environments often display different community types, each characterized by a unique composition of species (*Arumugam et al., 2011*; *Estrela et al., 2022*; *Gibson et al., 2016*; *Gonze et al., 2017*). Changes in environmental factors, fluctuations of species abundances, or transient invasions can trigger abrupt and irreversible transitions between two types (*Amor et al., 2020*; *Beardmore et al., 2018*; *Chaparro-Pedraza and de Roos, 2020*). Such transitions, also known as regime shifts, lead to the dramatic alterations of microbiome functions and are

tightly associated with host diseases, soil fertility, marine biogeochemical flux, or bioproduction yield (*Fuhrman et al., 2015*; *Hartmann and Six, 2023*; *Kumar and Hasty, 2023*; *Relman, 2012*; *Sun et al., 2023*). Understanding the origin of different community types has broad implications in environments, engineering, and health (*Gibson et al., 2016*; *Gonze et al., 2017*).

Multistability, a phenomenon where multiple stable states coexist for the same set of system parameters, is one of the most important mechanisms behind alternative community types and has been widely observed in a variety of natural or experimental microbiomes (*Amor et al., 2020*; *Dubinkina et al., 2019*; *Estrela et al., 2022*; *Fujita et al., 2023*; *Gonze et al., 2017*; *Khazaei et al., 2020*; *Mangal et al., 2023*; *Zaoli and Grilli, 2021*). Under perturbations, a multistable system can be driven away from a former stable state, cross the tipping point and reach a new steady state. The population can persist in the new state even after the perturbations are withdrawn, exhibiting a behavior of 'hysteresis' (*Gibson et al., 2016*; *Khazaei et al., 2020*). Despite its conceptual importance, the determinants underlying microbiota multistability have been poorly understood due to the intrinsic complexity of microbial interactions (*Dubinkina et al., 2019*).

Understanding the origin and the underlying determinants of multistability in complex microbiota is an important challenge in microbial ecology and evolution. In host-associated microbiota, regime shifts between alternative stable states can trigger the critical transitions between healthy microbiome and dysbiosis (*Fassarella et al., 2021*; *Relman, 2012*). Unravelling how different biological processes drive community multistability is important for the predictive control of complex microbiota, and can inspire novel treatments to modulate the communities towards the desired stable states or away from the unhealthy states.

Mathematical models are indispensable to decipher the emergent properties of complex populations (*van den Berg et al., 2022*). Different ecological models have been developed to analyze the origin of multistability in microbial communities. For instance, based on the generalized Lotka-Volterra (LV) model, Gonze D et al. showed that multistability could emerge in communities where different species mutually inhibited each other (*Gonze et al., 2017*). Dubinkina V et al. applied a consumer-resource model in which microbes competed for different metabolites, and showed that multistability required microbial species to have different stoichiometries of essential nutrients (*Dubinkina et al., 2019*). These works provide critical insights on the determinants of community multistability, yet the roles of many other mechanisms remain to be understood. Chief among these is horizontal gene transfer (HGT), a process where microbes share mobile genetic elements (MGEs) with their neighbors (*Soucy et al., 2015*).

Indeed, HGT is prevalent in microbial world and mobile genes occupy a substantial proportion in microbial gene pool (*Bordenstein and Reznikoff, 2005*; *Brito et al., 2016*; *Brockhurst et al., 2019*). For instance, among the closely related microbial isolates in the same human body site, HGT occurs in more than 40% of the species pairs (*Smillie et al., 2011*). In the draft genomes of human gut microbes, tens of thousands of genes are mobilizable (*Brito et al., 2016*). Depending on the encoded traits, MGEs shape microbial growth rates, host adaptability and even social interactions between microbes (*Rankin et al., 2011*). The dynamic gain and loss of MGEs are also a major source of evolutionary innovations of prokaryotic genomes (*Werren, 2011*). However, despite its functional and ecological importance, the interplay between genetic exchange and community multistability has not been rigorously examined in previous studies and a general conclusion remains lacking.

In this work, we focus on communities of competing microbes, a model system for microbiome multistability (*Dubinkina et al., 2019*; *Gilpin and Case, 1976*; *Gonze et al., 2017*). By combining mathematical modelling and numerical simulations, we demonstrated that the horizontal transfer of plasmids, a major type of MGEs, could promote the emergence of alternative stable states which otherwise would not exist. We further extended our analysis to scenarios where HGT changed interspecies interactions, where microbial communities were subjected to strong environmental selections and where microbes lived in metacommunities consisting of multiple local habitats. We also analyzed the role of different mechanisms, including interspecies interaction strength, the growth rate effects of MGEs, MGE epistasis and microbial death rates in shaping the multistability of microbial communities. These results created a comprehensive framework to understand how different dynamic processes, including but not limited to HGT rates, collectively shaped community multistability and diversity. Our results provide valuable insights for microbial ecology, health, and microbiome engineering.

## Results

### The role of HGT in bistability of two-species populations

To illustrate the basic concepts, let's first consider a population of two competing species. Without HGT, the community dynamics can be captured by the classic LV model that accounts for species growth rates ($\mu_1$ and $\mu_2$), interspecies interaction strengths ($\gamma_1$ and $\gamma_2$) and the dilution or death rate ($D$). When all the parameters are given, one can simulate the system dynamics from the initial abundances of the two species. The system will reach an equilibrium, characterized by the steady-state abundances of the two species. Bistability means the coexistence of two distinct stable states (*Figure 1A*; *Dubinkina et al., 2019*; *Gonze et al., 2017*). Depending on the initial abundances, either one of the two species can dominate the community. With bistability, even when the model parameters are not altered, a change in initial species abundances is sufficient to drive the system to a different stable state (*Figure 1A*). In contrast, if monostable, the population will always rest at the same steady state regardless of the initial composition (*Figure 1A*). Given the simplicity of this model, the condition of the system being bistable can be analytically solved as $\gamma_1 > \phi_2/\phi_1$ and $\gamma_2 > \phi_1/\phi_2$ where $\phi_1 = \frac{\mu_1 - D}{\mu_1}$ and $\phi_2 = \frac{\mu_2 - D}{\mu_2}$ (see Methods for more details). This condition predicted the thresholds on the strengths of interspecies interactions in order for the system to be bistable.

To analyze the effect of HGT on bistability, we applied a mathematical model that we previously developed to describe the conjugative transfer of plasmids between competing species (*Zhu et al., 2024*). While in classic LV model $\mu_1$ and $\mu_2$ are constants, our model dissected species growth rates into two components: the basal growth rates ($\mu_1^0$ and $\mu_2^0$) determined by chromosomal genes, and the growth rate effects ($\lambda_1$ and $\lambda_2$) of plasmids. The two components were assumed to combine multiplicatively: $\mu_1 = \mu_1^0 (1 + \lambda_1)$, $\mu_2 = \mu_2^0 (1 + \lambda_2)$. HGT creates subpopulations (denoted as $p_1$ and $p_2$) within each species that acquire the mobilizable genes from its competitor. The dynamics of $p_1$ and $p_2$ are governed by cell growth, plasmid transfer (the rate denoted as $\eta$) and plasmid loss (the rate denoted as $\kappa$). The gain and loss of $p_1$ and $p_2$ in turn lead to the temporal change of the effective growth rates ($\mu_1^e$ and $\mu_2^e$) of each species: $\mu_1^e = \mu_1 \left(1 + \lambda_2 \frac{p_1}{s_1}\right)$, $\mu_2^e = \mu_2 \left(1 + \lambda_1 \frac{p_2}{s_2}\right)$ (see Methods for more details). Therefore, plasmid transfer allows the two competitors to partially exchange their growth benefits or disadvantages. Here, we assumed that the species internal inhibition was independent of the acquired MGEs. We sought to understand whether the process of HGT would reshape the stability landscape of the system.

After HGT being introduced, analytical solution becomes infeasible due to the increase of modeling complexity. Instead, we determined whether the system was bistable by numerical simulations with randomized initial compositions (*Gilpin and Case, 1976*). In general, the resolution of this approach increases with the number of randomizations. Without loss of generality, here we randomly initialized the species abundances for 200 times between 0 and 1 following uniform distributions. Then we simulated the population dynamics for each initialization until steady state and obtained their species abundances. A system is called monostable if all populations starting from different compositions converge into the same state. In contrast, if two attractors exist, the system is called bistable (*Figure 1A*). We calculated the number of stable states for a wide range of $\mu_1$ and $\mu_2$ values and drew the phase diagram (*Figure 1B*). Whether it is easy for the system to achieve bistability is reflected by the total area of bistability region in the phase diagram. Our numerical results suggested that increasing HGT enlarged the region, creating bistability in many systems which would otherwise be monostable (*Figure 1B and C and D*). Here, the number of initializations was not critical for the conclusion (*Figure 1—figure supplement 1A*). The results were not fundamentally changed when the interspecies interaction strengths or gene transfer rates were altered (*Figure 1—figure supplement 1B-D*), suggesting the robustness of our conclusion.

Our results suggested that increasing HGT made it more feasible for the system to become bistable. However, this should not be taken as an assertion that the system will always be bistable when HGT exists (*Figure 1C and D*). For two species with large growth rate difference, the system might remain monostable when gene transfer rate increases (*Figure 1C*). In this case, the strong competitor will always win despite the gene flow. The effect of HGT in promoting bistability is more evident between species with smaller growth rate difference (*Figure 1B and D*).

Previous studies showed that the cell death rate determines the bistability of two interacting species (*Abreu et al., 2019*). To evaluate the impact of death rate $D$ on the interplay between HGT and system bistability, we performed additional analysis by calculating the bistability probability under

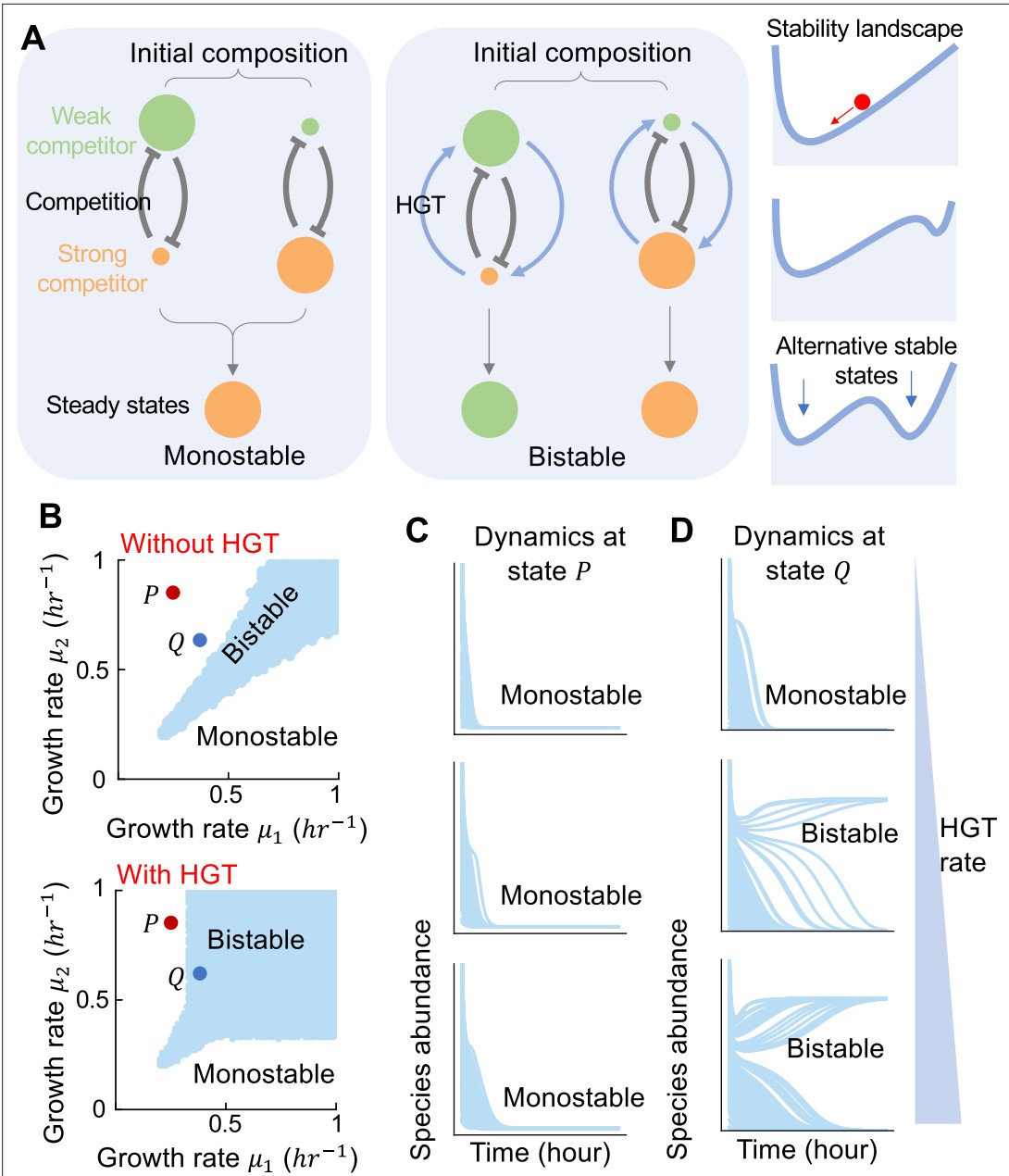

**Figure 1.** Horizontal gene transfer (HGT) promotes bistability of communities consisting of two competing microbes. (**A**) Schematics of monostable and bistable systems. A monostable system will always rest at the same steady state no matter where the system starts from. In contrast, a bistable system can reach two distinct steady states, depending on the initial ratio between the two competitors. The right panel is a diagram depicting how a microbial community responds to changes on initial species abundances. The red marble stands for the system state, which will always roll 'downhill' on the landscape towards stable states at the low points. The x axis represents the community composition and the valleys in the landscape are the stable states (or equilibrium points) where the community composition will stop changing. Depending on parameter values, the landscape can have one valley (called 'monostable') or two valleys (called 'bistable'). When the system is bistable, the marble can reach either valley, depending on where it starts. (**B**) The phase diagrams of the population with (bottom panel, $\eta = 0.2\,\mathrm{hr}^{-1}$) or without HGT (top panel). Different combinations of $\mu_1$ and $\mu_2$ between 0 and 1 $\mathrm{hr}^{-1}$ were tested. For each combination, the initial abundance of each species was randomized 200 times between 0 and 1 following uniform distributions. The system was monostable if all initializations led to the same steady state. Otherwise, the system was bistable. All combinations of $\mu_1$ and $\mu_2$ resulting in bistability were marked in blue in the diagram. Other parameters are $\gamma = 1.1$, $\kappa = 0.005\,\mathrm{hr}^{-1}$, $D = 0.2\,\mathrm{hr}^{-1}$, $\mu_1^0 = \mu_2^0 = 0.5\,\mathrm{hr}^{-1}$. Two states, $P$ ($\mu_1 = 0.3\,\mathrm{hr}^{-1}$, $\mu_2 = 0.8\,\mathrm{hr}^{-1}$) and $Q$ ($\mu_1 = 0.4\,\mathrm{hr}^{-1}$, $\mu_2 = 0.6\,\mathrm{hr}^{-1}$), were labeled as examples. (**C and D**) Population dynamics of $P$ and $Q$ under 100 different initializations. Here, the dynamic changes of species 1's abundance were shown. Three different HGT rates (0, 0.1, 0.2 $\mathrm{hr}^{-1}$ from top to bottom) were tested. When gene transfer rate increases, the system at $Q$ changed from monostable to bistable, while the system at $P$ remained monostable.

*Figure 1 continued on next page*

*Figure 1 continued*

The online version of this article includes the following figure supplement(s) for figure 1:

**Figure supplement 1.** The effect of HGT in two-species populations was robust against a variety of confounding factors.

**Figure supplement 2.** Dilution or death rate $D$ shapes the bistability of two-species communities.

**Figure supplement 3.** The strength of interspecies competitions influences the bistability of two competing species.

different $D$ values (*Figure 1—figure supplement 2*). Our results suggested that varying death rate indeed changed the bistability of the system. When the death rate equaled zero, MGEs that only modified growth rates would have no effects on population bistability. These results highlighted the importance of added death rate in driving multistability.

The strength of interspecies interactions also shaped the community bistability. By calculating the bistability probability under different values of $\gamma$ (*Figure 1—figure supplement 3*), we showed that while changing $\gamma$ didn't fundamentally alter our conclusion, the influence of HGT on population bistability became weak when interspecies interaction strength got smaller than 1. These results suggested that the role of HGT was more significant in populations with strong competitions.

## HGT shapes the stability landscape of multi-species communities

The modeling framework can be readily generalized to more complex communities consisting of multiple competing species by taking into account the gene transfer between every pair of species. Consider a community of $m$ species. Let $s_i$ be the abundance of the $i$-th species. $\gamma_{ij}$ represents the interspecies interaction strength that the $i$-th species imposes on the $j$-th species ($i, j = 1, 2, ..., m$). $\eta_{jki}$ is the transfer rate of the $s_j$-originated plasmid from species $k$ to species $i$. $\kappa_{ij}$ is the loss rate of the $j$-th plasmid in the $i$-th species. The growth rate effect of the $j$-th plasmid on the $i$-th species is denoted as $\lambda_{ij}$. $p_{ij}$ describes the abundance of the subpopulation in the $i$-th species that carries the plasmid from the $j$-th species. The effective growth rate of the $i$-th species can be calculated based on all the plasmids that it carries: $\mu_i^e = \mu_i \prod_{j \neq i} \left(1 + \lambda_{ij} \frac{p_{ij}}{s_i}\right)$. The population dynamics can then be simulated by $m + m^2$ equations that describe the temporal change of $s_i$ and $p_{ij}$, respectively (see Methods for more details).

To understand how changing HGT rate affects the stability landscape of complex microbiota, we estimated the number of alternative stable states by randomly initializing the abundance of each species 500 times between 0 and 1 following uniform distributions. For each initialization, we simulated the population dynamics for 2000 hrs till the systems reached equilibrium and calculated the steady-state abundances of different species (*Figure 2—figure supplement 1*). Limit cycles were not observed in our simulations (*Figure 2—figure supplement 2*).

To calculate the absolute number of stable states, we grouped the steady states into different attractors using a given threshold. Steady states with their Euclidean distances smaller than the threshold belonged to the same group (*Figure 2—figure supplement 3*). However, the estimation of this number can be biased by the intrinsic limitations of the numerical approach. For instance, the limited number of initializations can underestimate this number. In contrast, the inability of a system to reach equilibrium within the limited time window of simulations can cause its overestimation. To overcome such limitations, we developed an alternative metric $\chi$ to measure the multistability of a system by borrowing the concept of 'entropy'. Let $\varphi$ be the number of stable states and $x_i$ be the relative size of each domain of attraction ($\sum_i x_i = 1$, $i = 1, 2, \ldots, \varphi$). $\chi$ is calculated as $\chi = exp\left(-\sum_i x_i log x_i\right)$, which accounts for the absolute number of stable states as well as the size of each basin. $\chi$ equals 1 in monostable systems. $\chi$ gets larger when more stable states emerge or the distribution of the basin sizes become more even. Numerical tests suggested that $\chi$ was robust to the variations of distance threshold, simulation timespan or number of initializations (*Figure 2—figure supplement 4*).

As shown in *Figure 2*, when HGT is absent, the number of stable states was usually small in a multi-species population. Increasing gene transfer rate reshaped the stability landscape of a microbiome by creating additional stable states (*Figure 2A and B*). Each stable state is characterized by a different dominant species (*Figure 2A*). The number of stable states is positively associated with HGT rate (*Figure 2C*; *Figure 2—figure supplements 5 and 6*). This prediction is generally applicable to communities consisting of different number of species (*Figure 2—figure supplements 5 and 6*). For communities of two species, bistability is not possible when $\gamma_{ij}$'s are smaller than 1. However, for

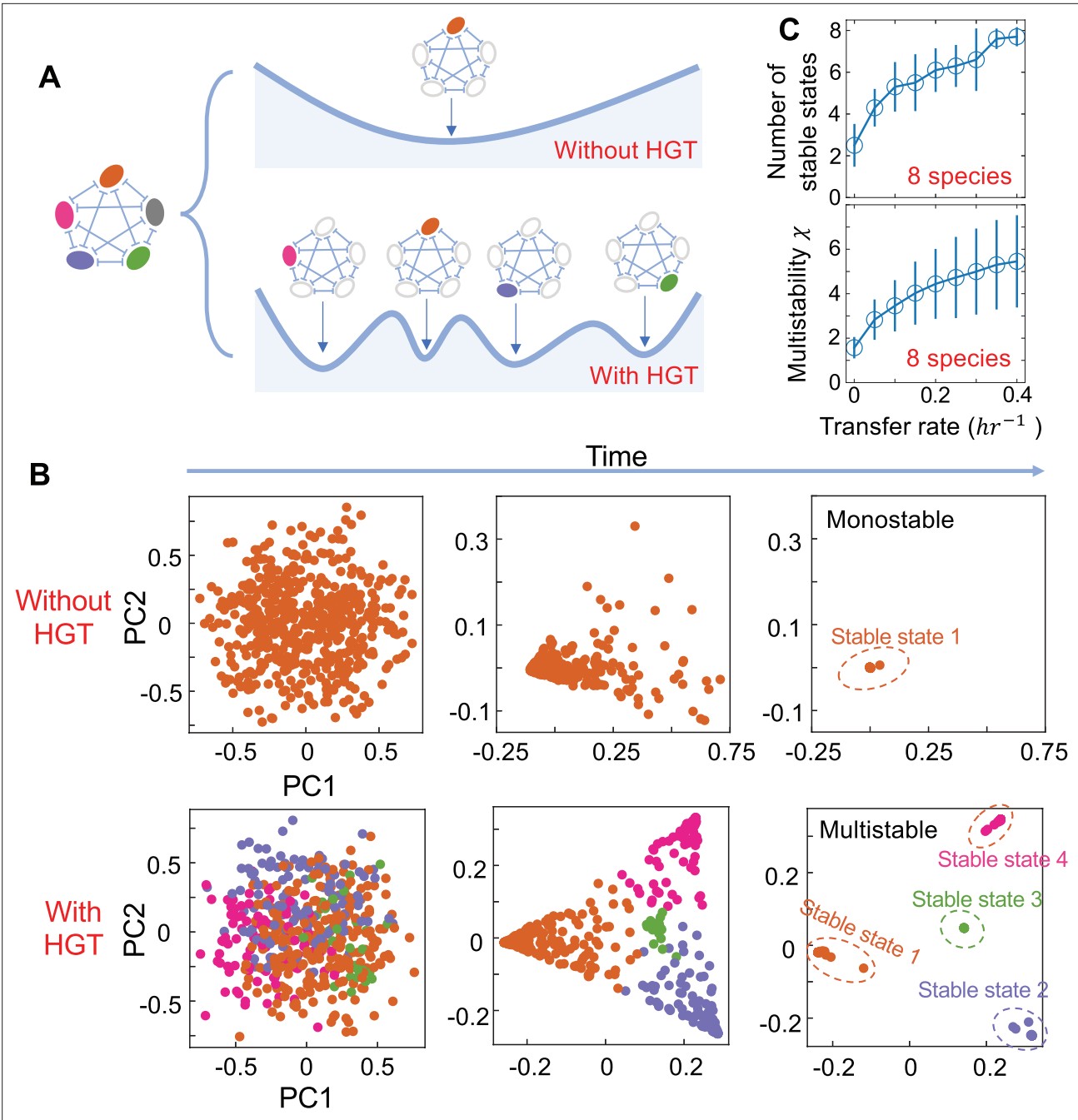

**Figure 2.** The number of alternative stable states in multi-species communities increases with HGT rate. (**A**) A schematic of the stability landscape of a population composed of multiple competing species with or without HGT. Each alternative stable state is characterized by a different dominant species (shown in filled circles). (**B**) The emergence of alternative stable states in a five-species community with gene transfer. We simulated the dynamics of 500 parallel communities, each with the same kinetic parameters but different initial compositions. The species compositions of these communities at three different timepoints were shown by principal component analysis from left to right. Without HGT ($\eta = 0$), all populations converged into a single stable state. In contrast, with HGT ($\eta = 0.4 \, \text{hr}^{-1}$), four different attractors appear. Here, populations reaching different stable states were marked with different colors. Other parameters are $\mu_1^0 = \mu_2^0 = \mu_3^0 = \mu_4^0 = \mu_5^0 = 0.5 \, \text{hr}^{-1}$, $\mu_1 = 0.52 \, \text{hr}^{-1}$, $\mu_2 = 0.48 \, \text{hr}^{-1}$, $\mu_3 = 0.44 \, \text{hr}^{-1}$, $\mu_4 = 0.60 \, \text{hr}^{-1}$, $\mu_5 = 0.31 \, \text{hr}^{-1}$, $\gamma = 1.05$, $\kappa = 0.005 \, \text{hr}^{-1}$, $D = 0.2 \, \text{hr}^{-1}$. (**C**) The number of stable states (top panel) and the multistability coefficient $\chi$ increase with HGT rate in communities consisting of eight competing species. We calculated the number of stable states by randomly initializing the species abundances 500 times between 0 and 1 following uniform distributions. Then we simulated the steady states and clustered them into different attractors by applying a threshold of 0.05 on their Euclidean distances. The data were presented as mean ± standard deviation of 10 replicates. Each replicate corresponded to a different combination of randomized species growth rates.

The online version of this article includes the following figure supplement(s) for figure 2:

*Figure 2 continued on next page*

*Figure 2 continued*

**Figure supplement 1.** Simulated dynamics of a 5-species community from time 0–2000 hr.

**Figure supplement 2.** No limit cycles were observed in simulated dynamics.

**Figure supplement 3.** The steady states were grouped into different attractors based on their Euclidean distances.

**Figure supplement 4.** The calculated multistability coefficient under different simulation conditions.

**Figure supplement 5.** Multistability of complex communities composed of different number of species.

**Figure supplement 6.** The relationship between community multistability and horizontal gene transfer in a broad range of HGT rate.

**Figure supplement 7.** For communities of five species, the number of stable states increases with HGT rate under different values of $\gamma_{ij}$.

**Figure supplement 8.** Initializing the $p_{ij}$ subpopulations with non-zero abundances didn't change the main conclusion.

**Figure supplement 9.** The number of stable states of five-species communities with heterogeneous inter-species interactions.

**Figure supplement 10.** The interplay between HGT and community multistability is not fundamentally changed by the range of growth rate effect $\lambda$.

**Figure supplement 11.** HGT promotes the multistability of communities with multiple niches.

communities of multiple species, multistability becomes possible even if all $\gamma_{ij}$ values are smaller than 1 (*Figure 2—figure supplement 7*). The effects of HGT on multistability held for different strengths of interspecies interactions (*Figure 2—figure supplement 7*). Initializing the $p_{ij}$ subpopulations with non-zero abundances did not change the conclusion, either (*Figure 2—figure supplement 8*).

Here we assumed the interactions among different species were homogeneous. To evaluate whether this assumption was critical, we extended our analysis by considering heterogenous interaction strengths in multispecies communities (*Figure 2—figure supplement 9*). In particular, we randomly sampled $\gamma_{ij}$ values from uniform distributions. Our results suggested the mean value and variance of $\gamma_{ij}$ played a role in shaping multistability. The effects of HGT on community multistability became stronger when the mean value of $\gamma_{ij}$ got larger than 1 and the variance of $\gamma_{ij}$ was small (*Figure 2—figure supplement 9*). Nevertheless, the heterogeneous distribution of $\gamma_{ij}$ didn't fundamentally change our conclusion.

MGEs can drive large growth rate differences when they encode adaptative traits like antibiotic resistance. In many cases, however, the growth rate effects of MGEs can be small. To evaluate the influence of $\lambda_{ij}$ magnitude, we also analyzed different ranges of growth rates effects of MGEs, by sampling $\lambda_{ij}$ values from uniform distributions with given widths (*Figure 2—figure supplement 10*). Greater width led to larger magnitude of growth rate effects. We used five-species populations as an example and tested different ranges. Our results suggested that multistability was more feasible when the growth rate effects of MGEs were small (*Figure 2—figure supplement 10B*). The qualitative relationship between HGT and community was not dependent on the range of growth rate effects (*Figure 2—figure supplement 10A*).

So far, our analysis has focused on communities where the interaction network is fully connected, and each species competes with every other member. In each of such populations, it is highly infeasible to achieve the stable coexistence of many species. While this analysis provides the basic proof of principle, natural environments often consist of many niches (*Finlay and Medzhitov, 2007*). Species living in the same niche compete while species from different niches can coexist (*Figure 2—figure supplement 11A*). To understand whether our conclusion is still applicable when niche effects are considered, we extended our analysis to communities carrying a random number of niches. Each species was randomly allocated into one of the niches, and the stability landscape was analyzed in the same way by randomly initializing the systems (see Methods for more details). Our results suggested that increasing HGT rate still promoted multistability, regardless of the niche structure of the communities (*Figure 2—figure supplement 11B-D*).

## The role of HGT in community multistability when MGEs modify interspecies interactions

Our previous analysis has focused on the role of HGT in communities where MGEs only affect species growth rates and have no influences on interspecies interactions. To evaluate how this modeling structure and the underlying assumptions affected the prediction, we extended our framework by accounting for the modifications of interspecies interactions by MGEs.

In nature, the horizontal transfer of some genes can change the interspecies interaction strengths. For instance, the sharing of many mobile genes can promote niche overlapping, leading to an increase of competition strength (*Bonham et al., 2017*; *Caro-Quintero and Konstantinidis, 2015*). To understand how the transfer of these genes would influence community multistability, we adapted the previous model by considering the dynamic change of competition strength during HGT (see Appendix 1 for more details). For two-species populations, we divided the interspecies interactions into two components: the basal competition strengths ($\gamma_1$ and $\gamma_2$) and the added parts by HGT ($\delta_1$ and $\delta_2$). The overall competition strength was calculated as $\gamma_1 + \delta_1 \frac{p_1}{s_1}$ and $\gamma_2 + \delta_2 \frac{p_2}{s_2}$, respectively. Positive $\delta$'s mean HGT promotes the interspecies competition, while negative $\delta$'s mean HGT reduces the strength of interspecies competition (*Figure 3A and B*). Numerical simulations with randomized parameters suggested that when $\delta$ was positive, HGT still promoted the bistability, whereas if $\delta$ was negative, gene transfer would reduce the chance of bistability (*Figure 3C and D*). We further extended our analysis to multispecies communities and calculated the number of stable states. As shown in *Figure 3E and F*, the number of stable states increased with HGT rate when MGEs promoted competition, while decreased with HGT rate when MGEs reduced competition.

These results suggested that the effects of HGT on bistability depended on how mobile genes shaped species growth rates and competitions. To place these two factors in a comprehensive framework, we further generalized the modeling structure, by accounting for the scenario where mobile genes modified growth rates and competitions at the same time. The effect of mobile genes on growth rates was represented by the magnitude of $\lambda$, and the influence on competition is described by the parameter $\delta$ (*Figure 3G, H and I*). By varying these two parameters, we can evaluate how the modeling structure and the underlying assumptions affected the baseline expectation. We performed additional simulations with broad ranges of $\lambda$ and $\delta$ values. In particular, we analyzed whether HGT would promote the likelihood of bistability in two-species communities compared with the scenario without gene transfer. Our results suggested that: (1) With or without HGT, reducing $\lambda$ (increasing ecological neutrality) promoted bistability (*Figure 3G and H*); (2) With HGT, increasing $\delta$ promoted bistability (*Figure 3H*); (2) Compared with the population without HGT, gene transfer promoted bistability when $\delta$ was zero or positive, while reduces bistability when $\delta$ was largely negative (*Figure 3I*). These results suggested that the interplay among HGT, species growth and interactions added new insights into the fundamental question of how microbes competing for a limited number of resources stably coexisted.

## The role of HGT in community multistability when MGE epitasis effects are present

In previous analysis, we also assumed that the effects of MGEs on host growth ($\lambda_1$ and $\lambda_2$) were independent of the host species. In nature, however, the same MGE can have different growth rate effects in different genomic backgrounds, a phenomenon called epistasis (*Acar Kirit et al., 2020*; *Gama et al., 2020*; *San Millan et al., 2014*; *Silva et al., 2011*). There are two types of epistasis: magnitude epistasis, where the host genomic background only affects the magnitude but not the sign of $\lambda$ value, and sign epistasis, where the same MGE is burdensome in one species while beneficial in the other host. The model can be readily extended to account for these two types of epistasis (see Appendix 2 for more details). In particular, we dissected the growth rates $\mu_1$ and $\mu_2$ into two components: the basal growth rate ($\mu_1^0$ and $\mu_2^0$), and the growth rate effects ($\lambda_{11}$ and $\lambda_{22}$) of the MGEs. The growth rates of $p_1$ and $p_2$ were calculated as $\mu_1 \left(1 + \lambda_{12}\right)$ and $\mu_2 \left(1 + \lambda_{21}\right)$, respectively. Here $\lambda_{ij}$ ($i, j = 1, 2$) stands for the growth rate effects of the $j$-th MGE in the $i$-th species. Without epistasis, the growth rate effects are independent of the host species ($\lambda_{11} = \lambda_{21}$ and $\lambda_{22} = \lambda_{12}$). The epistasis can be quantified by two ratios: $\xi_1 = \lambda_{21}/\lambda_{11}$ and $\xi_2 = \lambda_{12}/\lambda_{22}$. $\xi_1 > 0$ and $\xi_2 > 0$ represent magnitude epistasis, whereas $\xi_1 < 0$ or $\xi_2 < 0$ represent the sign epistasis. We carried out additional numerical simulations to evaluate how different epistasis types influenced the role of HGT in community stability. Our results suggested that with magnitude epistasis, HGT still enlarged the area of bistability region (*Figure 4A & B*). In contrast, sign epistasis overturned HGT's role on community stability: with sign epistasis, increasing gene transfer rate reduced the area of bistability region (*Figure 4C & D*). These results suggest that MGE epistasis might add another layer of complexities into the interplay between HGT and community stability.

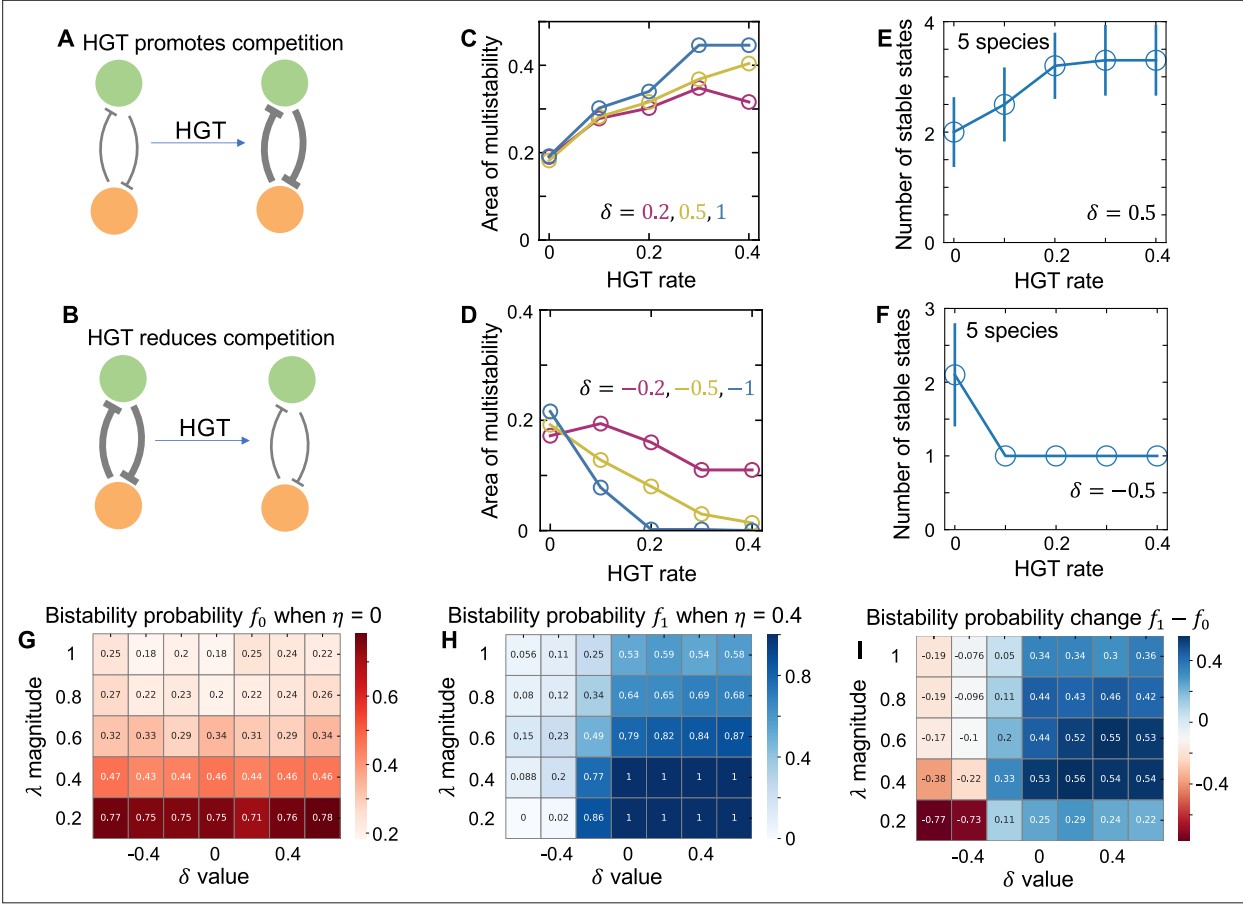

**Figure 3.** The effects of HGT on population multistability when MGEs promote or reduce the strength of interspecies competition. (**A and B**) The schematics of HGT promoting or reducing competition. (**C**) For populations of two species, when MGEs promote competition, increasing HGT rate enlarges the area of bistability region in the phase diagram. Here, $\delta$ describes the effect of mobile genes on the competition strength. Positive $\delta$ represents HGT promoting competition. In numerical simulations, we tested three different $\delta$ values (marked in different colors). When calculating the area of bistability region, we randomized $\mu_1$ and $\mu_2$ 500 times between 0 and 1 hr$^{-1}$ following uniform distributions while keeping $\mu_1^0$ and $\mu_2^0$ constants. For each pair of growth rates, we randomized the initial abundance of each species 200 times between 0 and 1 following uniform distributions. The system was monostable if all initializations led to the same steady state. Otherwise, the system was bistable. Then we calculated the fraction of growth rate combinations that generated bistability out of the 500 random combinations. Other parameters are $\gamma_1 = \gamma_2 = 1.1$, $\kappa = 0.005$ hr$^{-1}$, $D = 0.2$ hr$^{-1}$. (**D**) When MGEs reduce competition, the area of bistability region decreases with HGT rate. Three negative values $\delta$ were tested and shown as examples here. (**E**) For populations of five species, when MGEs promote competition, the number of stable states increases with HGT rate. We calculated the number of stable states by randomly initializing the species abundances 500 times. Then we simulated the steady states and clustered them into different attractors by applying a threshold of 0.05 on their Euclidean distances. The data were presented as mean ± standard deviation of 10 replicates. Each replicate corresponds to a different combination of randomized species growth rates. Other parameters are $\gamma_{i,j} = 1.1$, $\kappa = 0.005$ hr$^{-1}$, $D = 0.2$ hr$^{-1}$, $\delta = 0.5$. (**F**) When MGEs reduce competition, the number of stable states decrease with HGT rate. The data were presented as mean ± standard deviation of 10 replicates. $\delta = -0.5$ was used in the simulation. (**G**) For populations of two species without HGT, reducing the magnitude of $\lambda$ (promoting ecological neutrality) enlarges the area of bistability. When gene transfer rate equals zero, changing $\delta$ does not influence bistability. When calculating bistability probability, we randomized $\lambda_1$ and $\lambda_2$ 500 times between $-\alpha$ and $\alpha$ following uniform distributions. $\alpha$ represents the magnitude of $\lambda$ variations. $\mu_1$ and $\mu_2$ were calculated as $\mu_1^0 (1 + \lambda_1)$ and $\mu_2^0 (1 + \lambda_2)$, respectively. The overall competition strength was calculated as $\gamma_1 + \delta_1 \frac{p_1}{s_1}$ and $\gamma_2 + \delta_2 \frac{p_2}{s_2}$. We tested different combinations of $\alpha$ (the magnitude of $\lambda$ variations) and $\delta$ values. We then calculated the fraction ($f_0$) of growth rate combinations that generated bistability out of the 500 random combinations. Other parameters are $\gamma_1 = \gamma_2 = 1.1$, $\kappa = 0.005 h^{-1}$, $D = 0.2 h^{-1}$. (**H**) In populations of two species with HGT, reducing the magnitude of $\lambda$ (promoting ecological neutrality) or promoting $\delta$ value enlarges the area of bistability ($f_1$). $\eta$ being 0.4 was tested as an example here. (**I**) The effect of HGT on community bistability depends on how mobile genes modify growth rates and competition. Compared with the population without HGT, gene transfer promotes bistability when $\delta$ is zero or positive, while reduces bistability when $\delta$ is largely negative.

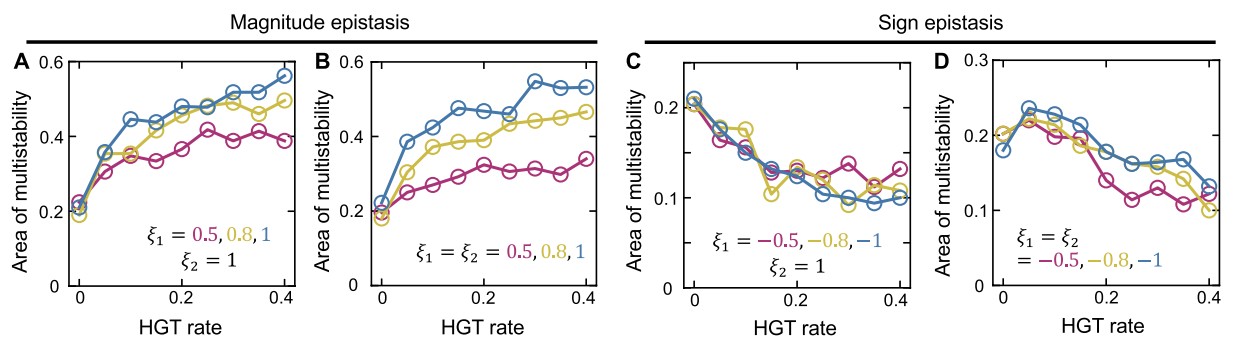

**Figure 4.** The influence of epistasis on the role of HGT in mediating population bistability. (**A and B**) With magnitude epistasis, increasing HGT rate still promotes bistability in populations of two competing species. (**C and D**) With sign epistasis, increasing HGT rate reduces the area of bistability region. For each type of epistasis, we considered two scenarios. In A and C, host genetic background influences the growth rate effect of only one MGE, while in B and D, host genetic background influences both MGEs. $\xi_1$ and $\xi_2$ are defined as $\lambda_{21}/\lambda_{11}$ and $\lambda_{12}/\lambda_{22}$, respectively. $\xi_1 > 0$ and $\xi_2 > 0$ represent magnitude epistasis, while $\xi_1 < 0$ or $\xi_2 < 0$ represents the sign epistasis. Other parameters are $\gamma_1 = \gamma_2 = 1.1$, $\kappa = 0.005\,\mathrm{hr}^{-1}$, $D = 0.2\,\mathrm{hr}^{-1}$.

## Interplay between HGT and the multistability of communities under strong selections

The growth rate effect of an MGE can be extreme in some cases. For instance, under strong antibiotic selection, only cells carrying the antibiotic resistance genes (ARGs) can survive. To examine whether our conclusion is still applicable in this scenario, we generalized our model by considering the transfer of an MGE in a population under strong environmental selection (see Methods for details). Specifically, we assumed that the MGE was initially carried by one species in the populations. Without HGT, only the donor species carrying the MGE can survive due to selection (*Figure 5A*). In this case, the system will always be monostable regardless of the initial species compositions. With HGT, the MGE can be transferred to other species, creating chances for other species to survive (*Figure 5B*). To understand how HGT would affect community stability under this condition, we first analyzed the phase diagram of the two-species population. Our simulation results suggested that bistability became possible when there existed gene transfer and increasing HGT rate in general enlarged the bistability area (*Figure 5C and D*). Here, without loss of generality we assumed that the MGE was initially carried only by species 1. Bistability was more feasible when $\mu_2 > \mu_1$ (*Figure 5C*), suggesting that the effect of HGT on promoting bistability would be stronger when the weak competitor acted as the MGE donor. We further analyzed the number of stable states in multispecies communities. As shown in *Figure 5E and F*, for these populations under antibiotic selection, increasing transfer rate of the MGE also promoted the emergence of multistability. This conclusion equally held under different interaction strengths or growth rate ranges (*Figure 5—figure supplement 1*), suggesting the general applicability of HGT's role in promoting multistability for communities under strong selection.

## HGT-enabled multistability promotes the regional coexistence of competing species

Our results highlight the role of HGT on shaping the stability landscape of closed local communities. Natural microbiomes, however, exist at scales broader than a localized population. A set of local communities in similar environments, also known as 'patches', form a metacommunity (*Leibold et al., 2004*; *Miller et al., 2018*). When multiple stable states coexist, the species compositions of different patches may differ from each other due to the variability of their initial configurations. Collectively, the multistability in the local patches might give rise to the regional species diversity in the metacommunity. This intuition leads to our hypothesis that HGT allows the stable coexistence of multiple competing species at regional scale, even when species exclusively outcompete each other in every local population.

To examine this hypothesis, we constructed metacommunities of 100 patches connected by dispersal of microbes. Without loss of generality, we considered a 2-D matrix of local patches and assumed that the dispersal only occurred between adjacent patches (*Figure 5—figure supplement 2A*, see Methods for more details). Each local patch carried 10 competing species that exchanged

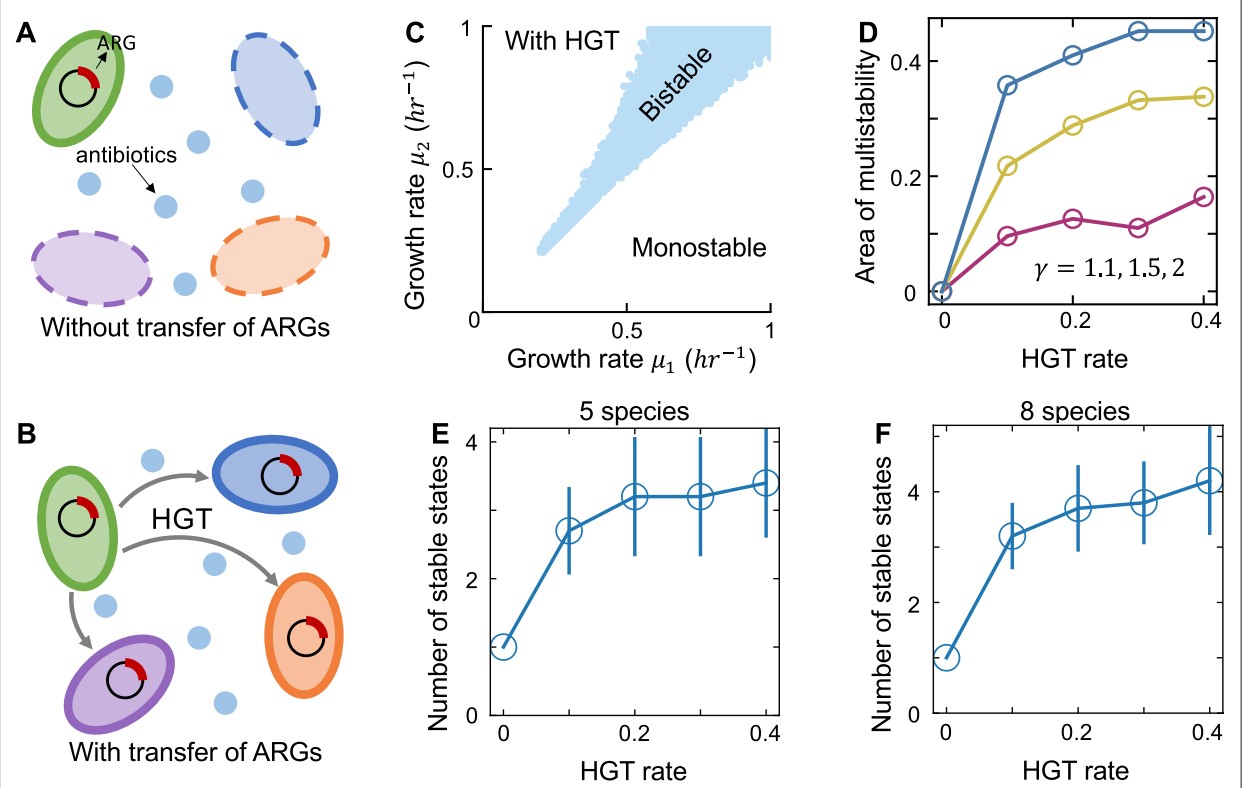

**Figure 5.** HGT creates chances of multistability for communities under strong environmental selection. (**A and B**) The schematics of communities under strong selection. Without HGT, only the donor species carrying the MGE can survive. HGT allows the other species to acquire MGE from the donor, creating opportunities for the other species to survive under strong selections. (**C**) The phase diagram of two-species populations transferring an MGE. Different combinations of $\mu_1$ and $\mu_2$ between 0 and 1 hr$^{-1}$ were tested. All combinations of $\mu_1$ and $\mu_2$ that led to bistability were marked in blue in the diagram. Other parameters are $\gamma = 1.1$, $\kappa = 0.005$ hr$^{-1}$, $D = 0.2$ hr$^{-1}$, $\mu_1^0 = \mu_2^0 = 0.5$ hr$^{-1}$. Here, we assumed that the ARG was initially carried only by species 1. Bistability was more feasible when $\mu_2 > \mu_1$. (**D**) Increasing HGT rate enlarges the area of bistability region in the phase diagram of two competing species transferring an MGE. Three different competition strengths were tested (marked in different colors). (**E and F**) For communities of 5 or 8 species under strong selection, increasing HGT rate promotes the emergence of alternative stable states. We calculated the number of stable states by randomly initializing the species abundances 500 times. Then we simulated the steady states and clustered them into different attractors. The carriage of the mobile gene changed the species growth rate from 0 to a positive value $\mu_i$. When calculating the number of stable states in multi-species communities, we randomly drew the $\mu_i$ values from a uniform distribution between 0.3 and 0.7 hr$^{-1}$. The data were presented as mean ± standard deviation of 10 replicates. Each replicate corresponds to a different combination of randomized species growth rates. Other parameters are $\gamma_{i,j} = 1.5$, $\kappa = 0.005$ hr$^{-1}$, $D = 0.2$ hr$^{-1}$.

The online version of this article includes the following figure supplement(s) for figure 5:

**Figure supplement 1.** For communities under antibiotic selection, the relationship between the number of stable states and HGT rate under different interactions strengths or growth rate ranges.

**Figure supplement 2.** HGT promotes the regional diversity in metacommunities of local populations connected by dispersal.

**Figure supplement 3.** The local multistability enables by HGT gives rise to regional diversity in a metacommunity without dispersal.

**Figure supplement 4.** HGT promotes the regional diversity in metacommunities regardless of the dispersal rate.

MGEs with each other. The dynamics of each patch was governed by the same set of parameters but initialized with different species compositions. We then numerically simulated the dynamics of the metacommunity till it reached the steady state.

Without HGT, the metacommunity exhibited strong spatial homogeneity at regional scale (*Figure 5—figure supplement 2B*). Different patches rested at similar species compositions. In contrast, HGT created a spatial pattern of species distributions across different patches, due to the emergent multistability (*Figure 5—figure supplement 2B*). To analyze the influence of HGT on regional diversity, we pooled all local populations together and calculated the collective species diversity in the pool using Shannon index. Our results suggested that while local diversity remained low regardless of HGT rate, the regional diversity of microbial species increased with HGT rate (*Figure 5—figure*

*supplement 2B and C*). The conclusion is applicable for different dispersal rates (*Figure 5—figure supplements 3 and 4*). These results highlight the fundamental role of HGT in community stability and diversity at local and regional scale.

## Discussion

Our work predicts that in many scenarios horizontal gene flow promotes multistability in communities of competing microbes. At the scale of local populations, the emergence of alternative stable states undermines the global stability of the community, making the system less resilient to perturbations and creating greater opportunities for regime shifts. In metacommunities, gene flow creates spatial heterogeneity of species compositions in different patches. Despite the low diversity in every local patch, HGT-driven multistability creates substantial diversity at regional scale. In microbial ecology, how microbes competing for a limited number of resources stably coexist has been a long-standing puzzle (*Grilli et al., 2017*; *Serván et al., 2018*). Our results highlight the importance of considering gene exchange when addressing this fundamental challenge.

While this work predicts the potential role of HGT in microbiota diversity and stability, several caveats need to be considered when applying this prediction. For MGEs with sign epistasis, or MGEs that relax interspecies competitions, HGT might reduce the number of stable states. The relative importance of gene flow in a specific community might also be context dependent. The effect of HGT in promoting multistability might be stronger between species with intense competition (*Figure 1—figure supplement 3*) and similar growth rates (*Figure 2—figure supplement 10*). The relative importance of HGT is also dependent on the maximum carrying capacity of the population. Simulations by scanning the $\eta$ value from $10^{-7}$ to 1 suggested that increasing HGT rates started to promote multistability when $\eta$ value exceeded $10^{-2}$ per hour (*Figure 2—figure supplement 6*). This corresponds to a conjugation efficiency of $10^{-11}$ cells. $hr^{-1}$.mL when the maximum carrying capacity equals $10^{9}$ cells. mL, or a conjugation efficiency of $10^{-14}$ cells. $hr^{-1}$.mL when the maximum carrying capacity equals $10^{12}$ cells. $mL^{-1}$. Therefore, in environments with high cell density and abundant MGEs, such as mammal gut, the role of HGT will be more prominent (*Liu et al., 2012*). In contrast, the influence of HGT on community multistability might become less important in populations with poor cell density and slow gene transfer.

Our work provided a comprehensive framework to explore how different parameters, including but not limited to HGT rates, collectively shaped community multistability. For instance, our simulations suggested that nonzero cell death or dilution rate was essential for the bistability of two competing species. Promoting competition strength in general enlarged multistability region. Besides, multistability was also dependent on community neutrality, that is the growth rate similarities among different competitors. Multistability is more feasible for communities with high neutrality. Which mechanisms contribute most to the multistability of a given community might be context-specific, depending on the values of different parameters in the population.

Our model allowed multiple plasmids to infect a single cell, but ignored the ecological interactions between different plasmids. These complex interactions are governed by many different processes: from systems that exclude the entry of other plasmids, incompatibility system that eliminates within-cell coexistence, to interference in the regulation of plasmid copy number (*Igler et al., 2022*; *Lipsitch et al., 2009*). These interactions can further change the distribution of plasmids among different species, alter the structure of gene transfer network, and reshape the stability landscape of a microbial community. However, the general conclusion regarding how these interactions shape population multistability remains to be established in future studies.

Previous studies documented the presence of multiple alternative community types in soil (*Fujita et al., 2023*), human vaginal, gut, and oral microbiomes (*Arumugam et al., 2011*; *Ding and Schloss, 2014*; *Ravel et al., 2011*). In particular, multiple groups of bacteria in human gut exhibit robust bistable abundance distributions, resulting in over 32 different stable state combinations (*Lahti et al., 2014*). Our results provide a plausible explanation for the large number of community types in these populations, even though our interpretation does not exclude other mechanisms including gradients of environmental factors (*Gonze et al., 2017*). Disentangling the contributions of HGT from the other factors in diverse natural microbiomes is a promising topic for future research.

Controlling the multistability of complex microbiota has important applications in human health. For instance, in gut microbiome, the development of unhealthy stable states is closely related to

many diseases such as obesity, type 2 diabetes mellitus and gastrointestinal disorders (*Canfora et al., 2019*; *Kootte et al., 2017*; *Moustafa et al., 2018*). Strategies have been developed to overcome the resilience of unhealthy states, including fecal transplantation or supplementation with probiotics (*Fassarella et al., 2021*). Our work predicts that the horizontal transfer of MGEs such as plasmids might be an additional mechanism for the emergence of the unhealthy states. Indeed, human gut is a 'hotspot' of gene flow (*Liu et al., 2012*). Previous studies suggested that some small molecules such as unsaturated fatty acids can inhibit the conjugative transfer of plasmids (*Fernandez-Lopez et al., 2005*; *Getino and de la Cruz, 2018*; *Getino et al., 2015*). By binding the type IV secretion traffic ATPase TrwD, these compounds limit the pilus biogenesis and DNA translocation (*Ripoll-Rozada et al., 2016*). Therefore, modulating MGE spread using these molecules might offer new opportunities to reshape the stability landscape and narrow down the attraction domains of the disease states (*Fernandez-Lopez et al., 2005*; *Palencia-Gándara et al., 2021*).

# Methods

## Key resources table

| Reagent type (species) or resource | Designation | Source or reference | Identifiers | Additional information |
|---|---|---|---|---|
| Software, algorithm | Codes for all numerical simulations | Github repository *Hong et al., 2025* | https://doi.org/10.5281/zenodo.14928352 | |

## Modeling framework of two competing species sharing mobile genes

We used a mathematical model developed in a previous work to analyze the population dynamics of two competing species transferring mobilizable plasmids with growth rate effects (*Zhu et al., 2024*). Briefly, the model includes four ordinary differential equations (ODEs):

$$\frac{dS_1}{dt} = \mu_1^e S_1 \left(1 - \frac{S_1 + R_2 S_2}{N_{m,1}}\right) - D S_1, \tag{1}$$

$$\frac{dS_2}{dt} = \mu_2^e S_2 \left(1 - \frac{S_2 + R_1 S_1}{N_{m,2}}\right) - D S_2, \tag{2}$$

$$\frac{dP_1}{dt} = \mu_1 \left(1 + \lambda_2\right) P_1 \left(1 - \frac{S_1 + R_2 S_2}{N_{m,1}}\right) + \eta_1^c \left(S_2 + P_1\right) \left(S_1 - P_1\right) - \left(D + \kappa_1\right) P_1, \tag{3}$$

$$\frac{dP_2}{dt} = \mu_2 \left(1 + \lambda_1\right) P_2 \left(1 - \frac{S_2 + R_1 S_1}{N_{m,2}}\right) + \eta_2^c \left(S_1 + P_2\right) \left(S_2 - P_2\right) - \left(D + \kappa_2\right) P_2. \tag{4}$$

$S_1$ and $S_2$ are the abundances of the two species. $R_1$ and $R_2$ describe the strengths of interspecies competition. $N_{m,1}$ and $N_{m,2}$ are the maximum carrying capacities for species 1 and 2, respectively. $D$ is the dilution rate. $P_i$ ($i = 1, 2$) represents the abundance of the subpopulation in the $i$-th species that acquires the mobilizable genes from its competitor. Here, we assumed that the populations would not lose their own native plasmids completely. The internal inhibition within each species was also assumed to be independent of the acquired MGEs. $\mu_i$ represents the maximum 'per capita' growth rate of the $i$-th species when interspecies HGT is absent. $\mu_i$ is calculated as $\mu_i = \mu_i^0 \left(1 + \lambda_i\right)$ where $\mu_i^0$ is the basal growth rate determined by the non-mobilizable genes and $\lambda_i$ is the growth rate effect of the plasmids. Interspecies gene flow allows each species to acquire plasmids from its competitor, resulting in the dynamic change of the effective species growth rates $\mu_i^e$, which is calculated as: $\mu_1^e = \mu_1 \left(1 + \lambda_2 \frac{P_1}{S_1}\right)$ and $\mu_2^e = \mu_2 \left(1 + \lambda_1 \frac{P_2}{S_2}\right)$. For instance, all cells of species 1 carry the plasmid 1. Therefore, their growth rate can be obtained as $\mu_1^0 \left(1 + \lambda_1\right)$. After HGT, the overall growth rate of species 1 changes because some of the individuals get the plasmid from the second species. Such a change is reflected by the $\lambda_2$ term in $\mu_1^e = \mu_1 \left(1 + \lambda_2 \frac{P_1}{S_1}\right)$. $\eta_1^c$ and $\eta_2^c$ stand for HGT rates. Here, we described the HGT process by mass action kinetics, an assumption commonly made for plasmid transfer dynamics. We further assumed that intraspecies and interspecies transfer occurred at the same rates. $\kappa_1$ and $\kappa_2$ represent the loss rates of the mobile genetic elements. For plasmids, $\kappa_1$ and $\kappa_2$ describe the loss rate by segregation errors.

Let $s_1 = S_1/N_{m,1}$, $s_2 = S_2/N_{m,2}$, $p_1 = P_1/N_{m,1}$, $p_2 = P_2/N_{m,2}$, $\gamma_2 = R_2 \frac{N_{m,2}}{N_{m,1}}$, $\gamma_1 = R_1 \frac{N_{m,1}}{N_{m,2}}$. The model can be further simplified as

$$\frac{ds_1}{dt} = \mu_1^e s_1 \left(1 - s_1 - \gamma_2 s_2\right) - D s_1,$$ (5)

$$\frac{ds_2}{dt} = \mu_2^e s_2 \left(1 - s_2 - \gamma_1 s_1\right) - D s_2,$$ (6)

$$\frac{dp_1}{dt} = \mu_1 \left(1 + \lambda_2\right) p_1 \left(1 - s_1 - \gamma_2 s_2\right) + \eta_1 \left(\varrho s_2 + p_1\right) \left(s_1 - p_1\right) - \left(D + \kappa_1\right) p_1,$$ (7)

$$\frac{dp_2}{dt} = \mu_2 \left(1 + \lambda_1\right) p_2 \left(1 - s_2 - \gamma_1 s_1\right) + \eta_2 \left(\varrho^{-1} s_1 + p_2\right) \left(s_2 - p_2\right) - \left(D + \kappa_2\right) p_2.$$ (8)

Here, $s_1$, $s_2$, $p_1$ and $p_2$ are dimensionless. $\eta_i$ is related with $\eta_i^c$ by $\eta_i = N_{m,i} \eta_i^c$. $\varrho$ describes the ratio between $N_{m,2}$ and $N_{m,1}$: $\varrho = N_{m,2}/N_{m,1}$. For simplicity, we assumed that the two species had the same carrying capacity, that is $\varrho = 1$. The unit of $\eta_1$ and $\eta_2$ is hour$^{-1}$. Without HGT, $p_1$ and $p_2$ become consistently 0, and the model is equivalent to the classic Lotka-Volterra (LV) framework:

$$\frac{ds_1}{dt} = \mu_1 s_1 \left(1 - s_1 - \gamma_2 s_2\right) - D s_1,$$ (9)

$$\frac{ds_2}{dt} = \mu_2 s_2 \left(1 - s_2 - \gamma_1 s_1\right) - D s_2.$$ (10)

## Stability analysis of the classic LV model

The classic LV model has four fixed points: $g_1 = [0,0]$, $g_2 = \left[0, 1 - \frac{D}{\mu_2}\right]$, $g_3 = \left[1 - \frac{D}{\mu_1}, 0\right]$, $g_4 = \left[\frac{\gamma_2 \mu_1 \mu_2 - D \gamma_2 \mu_1 - \mu_1 \mu_2 + D \mu_2}{\mu_1 \mu_2 (\gamma_1 \gamma_2 - 1)}, \frac{\gamma_1 \mu_1 \mu_2 - D \gamma_1 \mu_2 - \mu_1 \mu_2 + D \mu_1}{\mu_1 \mu_2 (\gamma_1 \gamma_2 - 1)}\right]$. The Jacobian matrix of each fixed point can be calculated as $J = \begin{bmatrix} \mu_1 - 2\mu_1 s_1^* - \mu_1 \gamma_2 s_2^* D & -\mu_1 \gamma_2 s_1^* \\ -\mu_2 \gamma_1 s_2^* & \mu_2 - 2\mu_2 s_2^* - \mu_2 \gamma_1 s_1^* - D \end{bmatrix}$, where $s_1^*$ and $s_2^*$ are the species abundances at each fixed point.

The system is bistable only when $g_2$ and $g_3$ are both stable. The Jacobian matrix of $g_2$ is $J_2 = \begin{bmatrix} \mu_1 - D - \mu_1 \gamma_2 + \frac{\mu_1 \gamma_2 D}{\mu_2} & 0 \\ -\mu_2 \gamma_1 + D \gamma_1 & D - \mu_2 \end{bmatrix}$. The eigenvalues of $J_2$ can be obtained as $\mu_1 - D - \mu_1 \gamma_2 + \frac{\mu_1 \gamma_2 D}{\mu_2}$ and $D - \mu_2$. $g_2$ is stable when both eigenvalues are negative, which leads to $\gamma_2 > \phi_1/\phi_2$ where $\phi_1 = \frac{\mu_1 - D}{\mu_1}$ and $\phi_2 = \frac{\mu_2 - D}{\mu_2}$. Similarly, $g_3$ is stable when $\gamma_1 > \phi_2/\phi_1$. Therefore, the condition for the system being bistable is $\gamma_1 > \phi_1/\phi_2$ and $\gamma_1 > \phi_2/\phi_1$.

## Bistability analysis of two-species populations by numerical simulations

We performed numerical simulations to determine the bistability of a population consisting of two competing species transferring mobile genes. Specifically, for a system with given parameters, we randomly initialized the species abundances between 0 and 1 following uniform distributions for 200 times. Then we simulated the dynamics of each population till the abundance of every species reached equilibrium, and we treated it as a steady state. We grouped the 200 steady states into different attractors by applying the threshold of 0.01. Two steady states with their Euclidean distance smaller than the threshold belonged to the same group. The Euclidean distance was calculated as $d_{ij} = \sqrt{\sum_{k=1}^{m} \left(s_{i,k} - s_{j,k}\right)^2}$ where $k$ is the species index and $s_{i,k}$ represents the abundance of the $k$-th species in the $i$-th steady state.

## Theoretical framework of multiple competing species transferring mobile genes

For a community of $m$ competing species that share mobile genes with each other, the modeling framework consists of two groups of ODEs:

$$\frac{ds_i}{dt} = \mu_i^e s_i \left(1 - s_i - \sum_{j \neq i} \gamma_{ji} s_j\right) - D s_i,$$ (11)

$$\frac{dp_{ij}}{dt} = \mu_i \left(1 + \lambda_{ij}\right) \left[\prod_{k \neq i,j} \left(1 + \lambda_{ik} \frac{p_{ik}}{s_i}\right)\right] p_{ij} \left(1 - s_i - \sum_{j \neq i} \gamma_{ji} s_j\right) + \left(s_i - p_{ij}\right) \sum_{k=1}^{n} \eta_{jki} p_{kj}$$
$$- \left(D + \kappa_{ij}\right) p_{ij}. \left(i \neq j\right).$$ (12)

$s_i$ is the abundance of the $i$-th species. $p_{ij}$ is the size of the subpopulation in the $i$-th species that acquires $s_j$-originated MGEs. $\gamma_{ji}$ is the negative interaction that $s_j$ imposes on the $i$-th species. $D$ is the dilution rate. $\mu_i^e$, the effective growth rate of $s_i$, is calculated by $\mu_i^e = \mu_i \prod_{j \neq i} \left(1 + \lambda_{ij} \frac{p_{ij}}{s_i}\right)$. $\mu_i$ is defined as $\mu_i = \mu_i^0 \left(1 + \lambda_{ii}\right)$ where $\mu_i^0$ is the basal growth rate, $\lambda_{ij}$ is the growth rate effect of the $s_j$-originated MGEs in the $i$-th species. $\eta_{jki}$ is the transfer rate of the $s_j$-originated plasmids from species $k$ to species $i$. $\kappa_{ij}$ is the loss rate of the $j$-th mobile genetic element in the $i$-th species.

## Analysis of multistability of complex communities

For a population of $m$ species with given niches and kinetic parameters, we initialized the system 500 times in the ranges of $0 < s_i < 1$ following uniform distributions. The initial conditions for $p_{ij}$'s are $p_{ii} = s_i$ and $p_{ij} = 0$ ($i \neq j$). Then the dynamics of each population was simulated for 2000 hr till the system reached the steady state. For each steady state, we obtained the corresponding species composition and grouped the 500 solutions into different stable states based on their Euclidean distances.

## Horizontal gene transfer within communities under strong environmental selection

Under strong antibiotic selections, only cells carrying antibiotic resistant MGEs can survive. To examine whether our conclusion is still applicable in this scenario, we generalized our model by considering the transfer of an antibiotic-resistant gene (ARG) in a population of $m$ species under strong antibiotic selection. Within each species, only cells carrying the MGE can grow, with $\mu_i$ being their maximum growth rate. The cells without the MGE will be eventually depleted by dilution $D$. The community dynamics can be described by the following ODEs ($i = 1, 2, \ldots, m$):

$$\frac{ds_i}{dt} = \mu_i p_i \left(1 - s_i - \sum_{j \neq i} \gamma_{ji} s_j\right) - D s_i, \tag{13}$$

$$\frac{dp_i}{dt} = \mu_i p_i \left(1 - s_i - \sum_{j \neq i} \gamma_{ji} s_j\right) + \left(s_i - p_i\right) \sum_{j=1}^{m} \eta_{ji} p_j - \kappa_i p_i - D p_i. \tag{14}$$

$s_i$ is the total abundance of the $i$-th species, and $p_i$ is the abundance of cells carrying the MGE in $s_i$. In the first equation, the first term describes the overall growth of the $i$-th species, which is only contributed by $p_i$ since the MGE-free cells are not able to grow. $\eta_{ji}$ is the MGE transfer rate from the $j$-th to the $i$-th species. $\kappa_i$ is MGE loss rate in the $i$-th species. $\gamma_{ji}$ describes the inter-species competition strength.

Without loss of generality, we assumed the species 1 was the MGE donor and $p_1$ was equal with $s_1$. Without HGT, the MGE only resides in the donor species, while HGT allows the MGE to be shared with other species. To understand how HGT rate influences community multistability, we calculated the phase diagram of populations of two competing species. Increasing HGT rate enlarged the bistability region. We further analyzed the number of alternative stable states in communities of 5 or 8 species. In particular, we randomly initialized the species abundances 500 times, simulated the steady states and clustered them into different attractors. When calculating the number of stable states in multi-species communities, we randomly drew the $\mu_i$ values from a uniform distribution between 0.3 and 0.7 hr⁻¹ Our results suggested that the number of stable states in general increased with HGT rate.

## Niche-based model of multi-species communities

For a community consisting of $m$ competing species, let the number of niches be $l$. Each niche has a maximum carrying capacity $N_m^{(i)}$ ($i = 1, 2, \ldots, l$). Each species lives in one of the niches. Let $M$ be the map from species index to the index of its niche. $M_i = k$ means that the $i$-th species lives in the $k$-th niche. For species $i$ and $j$, if $M_i = M_j$, these two species belong to the same niche and compete with each other. Species from different niche do not directly interact. Therefore, the model becomes:

$$\frac{ds_i}{dt} = \mu_i^e s_i \left(1 - \frac{s_i + \sum_{\{j : M_j = M_i\}} \gamma_{ji} s_j}{N_m^{(M_i)}}\right) - D s_i, \tag{15}$$

$$\frac{dp_{ij}}{dt} = \mu_i \left(1 + \lambda_{ij}\right) \left[\prod_{k \neq i,j} \left(1 + \lambda_{ik} \frac{p_{ik}}{s_i}\right)\right] P_{ij} \left(1 - \frac{s_i + \sum_{\{j: M_j = M_i\}} \gamma_{ji} s_j}{N_m^{(M_i)}}\right) + \left(s_i - p_{ij}\right) \sum_{k=1}^n \eta_{jki} p_{kj} \quad (16)$$
$$- \left(D + \kappa_{ij}\right) p_{ij}. \, \left(i \neq j\right).$$

Here, $s_i$ is the abundance of the $i$-th species. $p_{ij}$ is the size of the subpopulation in the $i$-th species that acquires $s_j$-originated mobile genes. $\gamma_{ji}$ is the negative interaction that $s_j$ imposes on the $i$-th species. $\eta_{jki}$ is the transfer rate of the $s_j$-originated genes from species $k$ to species $i$.In simulations, the number of niches was randomized between 2 and $m/2$. The maximum carrying capacity of each niche was randomized between 0 and 1 following uniform distributions. Then each species was randomly allocated into one of the niches.

## Theoretical framework of metacommunity dynamics

Consider a metacommunity composed of $u \times v$ local patches. We assumed a grid-like arrangement of patched connected by dispersal terms $H$. The location of each patch can be described by two coordinates $[a, b]$ ($a = 1, 2, \ldots, u$ and $b = 1, 2, \ldots, v$). Let $\omega$ be the dispersal rate between the adjacent patches. The dynamics of the metacommunity can then be described by generalizing *Equations 11 and 12* to each local patch:

$$\frac{ds_{a,b,i}}{dt} = \mu_{a,b,i}^e s_{a,b,i} \left(1 - s_{a,b,i} - \sum_{j \neq i} \gamma_{ji} s_{a,b,j}\right) - D s_{a,b,i} + H_s^{(a,b)}, \quad (17)$$

$$\frac{dp_{a,b,i,j}}{dt} = \mu_i \left(1 + \lambda_{ij}\right) \left[\prod_{k \neq i,j} \left(1 + \lambda_{ik} \frac{p_{a,b,i,k}}{s_{a,b,i}}\right)\right] p_{a,b,i,j} \left(1 - s_{a,b,i} - \sum_{j \neq i} \gamma_{ji} s_{a,b,j}\right) \quad (18)$$
$$+ \left(s_{a,b,i} - p_{a,b,i,j}\right) \sum_{k=1}^n \eta_{jki} p_{a,b,k,j} - \left(D + \kappa_{ij}\right) p_{a,b,i,j} + H_p^{(a,b)}. \, \left(i \neq j\right).$$

For the patch located at $[a, b]$, $s_{a,b,i}$ is the abundance of the $i$-th species, and $p_{a,b,i,j}$ is the size of the subpopulation in the $i$-th species that acquires $s_j$-originated mobile genes. The effective growth rate $\mu_{a,b,i}^e$ is calculated as $\mu_{a,b,i}^e = \mu_i \prod_{j \neq i} \left(1 + \lambda_{ij} \frac{p_{a,b,i,j}}{s_{a,b,i}}\right)$.

$H_s^{(a,b)}$ and $H_p^{(a,b)}$ are the gains of $s_{a,b,i}$ and $p_{a,b,i,j}$ by dispersal, respectively. We assumed that microbial dispersal only occurred between adjacent patches. Therefore, $H_s^{(a,b)}$ was calculated by summing the dispersal terms at four directions: left, right, up, and down. The left component can be calculated as

$$H_{s,left}^{(a,b)} = \begin{cases} \omega \left(s_{a-1,b,i} - s_{a,b,i}\right), if \, a - 1 > 0 \\ 0, if \, a - 1 \leq 0 \end{cases}. \quad (19)$$

Similarly, $H_{s,right}^{(a,b)}$, $H_{s,up}^{(a,b)}$ and $H_{s,down}^{(a,b)}$ is calculated as:

$$H_{s,right}^{(a,b)} = \begin{cases} \omega \left(s_{a+1,b,i} - s_{a,b,i}\right), if \, a + 1 < u + 1 \\ 0, if \, a + 1 \geq u + 1 \end{cases}, \quad (20)$$

$$H_{s,up}^{(a,b)} = \begin{cases} \omega \left(s_{a,b+1,i} - s_{a,b,i}\right), if \, b + 1 < v + 1 \\ 0, if \, b + 1 \geq v + 1, \end{cases} \quad (21)$$

$$H_{s,down}^{(a,b)} = \begin{cases} \omega \left(s_{a,b-1,i} - s_{a,b,i}\right), if \, b - 1 > 0 \\ 0, if \, b - 1 \leq 0 \end{cases}. \quad (22)$$

$H_s^{(a,b)}$ can then be obtained as $H_{s,left}^{(a,b)} + H_{s,right}^{(a,b)} + H_{s,up}^{(a,b)} + H_{s,down}^{(a,b)}$. $H_p^{(a,b)}$ can be formulated in the similar way.

## The collective species diversity in a metacommunity

To analyze the total species diversity in a metacommunity, we first calculated the total abundance of each species across all patches: $s_i = \sum_{a=1,2,\ldots,u} \sum_{b=1,2,\ldots,v} s_{a,b,i}$. Then we measured the diversity using Shannon index, a common metric used in ecology. Shannon index can be calculated as $exp\left[ -\sum_{i=1}^{m} \frac{s_i}{s_T} log \frac{s_i}{s_T} \right]$. $s_T$ is the total species abundance: $s_T = \sum_{i=1}^{m} s_i$.

## Biological relevance of our analysis

The empirically measured gene transfer rates (denoted as $\eta^c$, with the unit of $cells^{-1}.mL.hour^{-1}$) should be multiplied by $N_m$, the maximum carrying capacity of the population, before being used in our model. Therefore, the transfer rate $\eta$ in our analysis are several orders of magnitude higher than the experimentally measured rates (*Kosterlitz et al., 2022*; *Lopatkin et al., 2017*). In this work, we focused on HGT rate in the range of $0 < \eta_i < 0.4\,hour^{-1}$. Whether this range is biologically relevant depends on $N_m$ of the population. Considering the typical estimates of plasmid transfer rates cross species ($10^{-13}$ $cells^{-1}.mL.hour^{-1}$ from *Klebsiella pneumoniae* to *Escherichia coli*) and within species ($10^{-7}$ $cells^{-1}.mL.hour^{-1}$, between *E. coli* strains) provided by a previous study (*Kosterlitz et al., 2022*), HGT rates in our analysis are relevant in various environments from human colon (~$10^{12}$ cells per gram; *Dieterich et al., 2018*) to deep oceanic subsurface ($10^6$~$10^8$ cells per gram; *Flemming and Wuertz, 2019*).

## Acknowledgements

This study was partially supported the National Natural Science Foundation of China (12401660 to TW) and Shenzhen Institute of Synthetic Biology Scientific Research Program (HSE499011086 to TW).

## Additional information

### Funding

| Funder | Grant reference number | Author |
| --- | --- | --- |
| National Natural Science Foundation of China | 12401660 | Teng Wang |
| Shenzhen Institue of Synthetic Biology Scientific Research Program | HSE499011086 | Teng Wang |

The funders had no role in study design, data collection and interpretation, or the decision to submit the work for publication.

### Author contributions

Juken Hong, Formal analysis, Validation, Investigation, Visualization, Methodology, Writing – original draft; Wenzhi Xue, Investigation, Visualization, Writing – original draft; Teng Wang, Conceptualization, Resources, Formal analysis, Supervision, Funding acquisition, Validation, Investigation, Visualization, Methodology, Writing – original draft, Writing – review and editing

### Author ORCIDs

Teng Wang https://orcid.org/0009-0000-5176-5095

Reviewer #2 (Public review): https://doi.org/10.7554/eLife.99593.3.sa1
Reviewer #3 (Public review): https://doi.org/10.7554/eLife.99593.3.sa2
Author response https://doi.org/10.7554/eLife.99593.3.sa3

## Additional files

### Supplementary files
MDAR checklist

## Data availability

All the codes and source data associated with the numerical simulations and analysis of this manuscript are available on GitHub (copy archived at *Hong et al., 2025*).

The following dataset was generated:

| Author(s) | Year | Dataset title | Dataset URL | Database and Identifier |
|---|---|---|---|---|
| Hong J, Xue W, Wang T | 2025 | MultistabilityByHGT | https://doi.org/10.5281/zenodo.14928352 | Zenodo, 10.5281/zenodo.14928352 |

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

# Appendix 1

## The effects of HGT on population multistability when MGEs change interspecies competitions

In nature, the HGT of some genes can change the competition strengths between different species. To understand how the transfer of these genes would influence community multistability, we adapted the main model, by accounting for the dynamic change of competition strength during gene transfer. For a population of two competing species, the model consists of four ODEs:

$$\frac{ds_1}{dt} = \mu_1 s_1 \left[ 1 - s_1 - \left( \gamma_2 + \delta_2 \frac{p_2}{s_2} \right) s_2 \right] - D s_1,$$

$$\frac{ds_2}{dt} = \mu_2 s_2 \left[ 1 - s_2 - \left( \gamma_1 + \delta_1 \frac{p_1}{s_1} \right) s_1 \right] - D s_2,$$

$$\frac{dp_1}{dt} = \mu_1 p_1 \left[ 1 - s_1 - \left( \gamma_2 + \delta_2 \frac{p_2}{s_2} \right) s_2 \right] + \eta_1 \left( s_2 + p_1 \right) \left( s_1 - p_1 \right) - \left( D + \kappa_1 \right) p_1,$$

$$\frac{dp_2}{dt} = \mu_2 p_2 \left[ 1 - s_2 - \left( \gamma_1 + \delta_1 \frac{p_1}{s_1} \right) s_1 \right] + \eta_2 \left( s_1 + p_2 \right) \left( s_2 - p_2 \right) - \left( D + \kappa_2 \right) p_2.$$

Here, we assumed that mobile genes didn't cause growth burden or benefits on the species growth rates. The interspecies competitions are determined by two components: the basal competition strengths ($\gamma_1$ and $\gamma_2$) and the added parts by HGT ($\delta_1$ and $\delta_2$). The overall competition strengths are calculated as $\gamma_1 + \delta_1 \frac{p_1}{s_1}$ and $\gamma_2 + \delta_2 \frac{p_2}{s_2}$, respectively. Positive $\delta$ values represent HGT promoting competition, while negative $\delta$ values represent HGT reducing competition.

This framework can be extended to populations composed of multiple species. For a community of $m$ species. The population dynamics can be described by $2m$ ODEs:

$$\frac{ds_i}{dt} = \mu_i s_i [1 - s_i - \sum_{j \neq 1} (\gamma_{ji} + \delta_{ji} \frac{p_{ji}}{s_j} s_j) - D s_i],$$

$$\frac{dp_{ij}}{dt} = \mu_i p_{ij} [1 - s_i - \sum_{j \neq i} (\gamma_{ji} + \delta_{ji} \frac{p_{ji}}{s_j}) s_j] + (s_i - p_{ij}) \sum_{k=1}^{m} \eta_{jki} p_{kj} - (D + \kappa_{ij}) p_{ij}. \, (i \neq j).$$

$s_i$ represents the abundance of the $i$-th species, and $p_{ij}$ represents the abundance of cells in the $i$-th species that acquires $s_j$-originated mobile genetic elements. We assumed $p_{ii} = s_i$. $\mu_i$ is the maximum growth rate of $s_i$. $\gamma_{ji}$ stands for the basal interaction that $s_j$ imposes on the $i$-th species without HGT. $\delta_{ji}$ describes the change of inter-species competition due to gene transfer. The overall interaction that $s_j$ imposes on the $i$-th species becomes $\gamma_{ji} + \delta_{ji} \frac{p_{ji}}{s_j}$. $\eta_{jki}$ is the transfer rate of the $s_j$-originated MGEs from species $k$ to species $i$. $D$ and $\kappa_{ij}$ are the dilution and MGE loss rate, respectively.

How HGT affects bistability depends on the influences of mobile genes on species growth rates and competitions. To place these two factors in a comprehensive framework, we further generalized the model structure, by accounting for the scenario where MGEs simultaneously modify growth rates and competitions:

$$\frac{ds_1}{dt} = \mu_1^e s_1 \left[ 1 - s_1 - \left( \gamma_2 + \delta_2 \frac{p_2}{s_2} \right) s_2 \right] - D s_1,$$

$$\frac{ds_2}{dt} = \mu_2^e s_2 \left[ 1 - s_2 - \left( \gamma_1 + \delta_1 \frac{p_1}{s_1} \right) s_1 \right] - D s_2,$$

$$\frac{dp_1}{dt} = \mu_1 \left( 1 + \lambda_2 \right) p_1 \left[ 1 - s_1 - \left( \gamma_2 + \delta_2 \frac{p_2}{s_2} \right) s_2 \right] + \eta_1 \left( s_2 + p_1 \right) \left( s_1 - p_1 \right) - \left( D + \kappa_1 \right) p_1,$$

$$\frac{dp_2}{dt} = \mu_2 \left( 1 + \lambda_1 \right) p_2 \left[ 1 - s_2 - \left( \gamma_1 + \delta_1 \frac{p_1}{s_1} \right) s_1 \right] + \eta_2 \left( s_1 + p_2 \right) \left( s_2 - p_2 \right) - \left( D + \kappa_2 \right) p_2.$$

Here, $\mu_1$ and $\mu_2$ are calculated as $\mu_1 = \mu_1^0 \left(1 + \lambda_1\right)$ and $\mu_2 = \mu_2^0 \left(1 + \lambda_2\right)$, respectively. $\mu_1^0$ and $\mu_2^0$ are the basal growth rates determined by the non-mobilizable genes. $\lambda_1$ and $\lambda_2$ are the growth rate effects of the MGEs. The effective species growth rates are calculated as: $\mu_1^e = \mu_1 \left(1 + \lambda_2 \frac{P_1}{S_1}\right)$ and $\mu_2^e = \mu_2 \left(1 + \lambda_1 \frac{P_2}{S_2}\right)$. By changing the $\delta$ values and the magnitudes of $\lambda$'s, this model structure allowed us to evaluate how the interplay among HGT, species growth and interactions affected the baseline expectation.

## Appendix 2

### Modeling the epistasis of mobile genetic elements

To evaluate the influence of MGE epistasis on the role of HGT in mediating population bistability, we dissected the growth rates $\mu_1$ and $\mu_2$ into two components: the basal growth rate ($\mu_1^0$ and $\mu_2^0$), and the growth rate effects ($\lambda_{11}$ and $\lambda_{22}$) of the mobilizable genes. The growth rates of $p_1$ and $p_2$ were calculated as $\mu_1(1 + \lambda_{12})$ and $\mu_2(1 + \lambda_{21})$, respectively. Here, $\lambda_{ij}$ $(i, j = 1, 2)$ describes the growth rate effects of the $j$-th mobile gene in the $i$-th species. Without epistasis ($\lambda_{11} = \lambda_{21}$ and $\lambda_{22} = \lambda_{12}$), the growth rate effects are independent of host species. Epistasis leads to the difference between $\lambda_{11}$ and $\lambda_{21}$ (or between $\lambda_{22}$ and $\lambda_{12}$). The population dynamics can be described by the following ODEs:

$$\frac{ds_1}{dt} = \mu_1 \left(1 + \lambda_{12} \frac{p_1}{s_1}\right) s_1 \left(1 - s_1 - \gamma_2 s_2\right) - D s_1,$$

$$\frac{ds_2}{dt} = \mu_2 \left(1 + \lambda_{21} \frac{p_2}{s_2}\right) s_2 \left(1 - s_2 - \gamma_1 s_1\right) - D s_2,$$

$$\frac{dp_1}{dt} = \mu_1 \left(1 + \lambda_{12}\right) p_1 \left(1 - s_1 - \gamma_2 s_2\right) + \eta_1 \left(s_2 + p_1\right) \left(s_1 - p_1\right) - \left(D + \kappa_1\right) p_1,$$

$$\frac{dp_2}{dt} = \mu_2 \left(1 + \lambda_{21}\right) p_2 \left(1 - s_2 - \gamma_1 s_1\right) + \eta_2 \left(s_1 + p_2\right) \left(s_2 - p_2\right) - \left(D + \kappa_2\right) p_2.$$

Here, $\gamma_1$ and $\gamma_2$ are the competition strengths. $D$ is the dilution rate. $\eta_1$ and $\eta_2$ are transfer rates. $\kappa_1$ and $\kappa_2$ are the loss rates of the mobilizable genes.

We quantified the epistasis using by two ratios: $\xi_1 = \lambda_{21}/\lambda_{11}$ and $\xi_2 = \lambda_{12}/\lambda_{22}$. $\xi_1 > 0$ and $\xi_2 > 0$ represent magnitude epistasis, while $\xi_1 < 0$ or $\xi_2 < 0$ represent the sign epistasis. In numerical simulations, we tested both types of epistasis. Specifically, for systems of two competing species, we approximated the areas of bistability regions under different $\xi_1$ and $\xi_2$ values, by numerical simulations with randomized parameters. Given competition strength, we randomized $\mu_1$ and $\mu_2$ 500 times between 0 and 1 hr$^{-1}$ following uniform distributions while keeping $\mu_1^0$ and $\mu_2^0$ constants. For each pair of growth rates, we randomized the initial abundance of each species 200 times between 0 and 1 following uniform distributions. The system was monostable if all initializations led to the same steady state. Otherwise, the system was bistable. Then we calculated the fraction of growth rate combinations that generate bistability out of the 500 random combinations.

