## [Editor Report · eLife Assessment]

This manuscript offers **valuable** theoretical predictions on how horizontal gene transfer (HGT) can lead to alternative stable states in microbial communities. Using a modeling framework, **solid** theoretical evidence is provided to support the claimed role of HGT. However, given that the model has many degrees of freedom, a more comprehensive analysis of the role of different parameters could strengthen the study. Additionally, potential interactions between plasmids that carry out HGT are not discussed in the model. This paper would be of interest to researchers in microbiology, ecology, and evolutionary biology.

---

## [Referee Report · Reviewer #2 (Public review)]

Summary:

In this work, the authors use a theoretical model to study the potential impact of Horizontal Gene Transfer on the number of alternative stable states of microbial communities. For this, they use a modified version of the competitive Lotka Volterra model-which accounts for the effects of pairwise, competitive interactions on species growth-that incorporates terms for the effects of both an added death (dilution) rate acting on all species and the rates of horizontal transfer of mobile genetic elements-which can, in turn, affect species growth rates. The authors analyze the impact of horizontal gene transfer in different scenarios--such as bistability between pairs of species and multistability in communities--over an extended range of parameter values. In almost all these cases, the authors report an increase in either the number of alternative stable states or the parameter region (e.g. growth rate values) in which they occur.

Understanding the origin of alternative stable states in microbial communities and how often they may occur is an important challenge in microbial ecology and evolution. Shifts between these alternative stable states can drive transitions between e.g. a healthy microbiome and dysbiosis. A better understanding of how horizontal gene transfer can drive multistability could help predict alternative stable states in microbial communities, as well as inspire novel treatments to steer communities towards the most desired (e.g. healthy) stable states. In my opinion, this manuscript is a solid theoretical approach to the subject.

Strengths:

- Generality of the model: the work is based on a phenomenological model that has been extensively used to predict the dynamics of ecological communities in many different scenarios.

- The question of how horizontal gene transfer can drive alternative stable states in microbial communities is important and there are very few studies addressing it.

Weaknesses:

- In the revised version of the manuscript, the authors significantly extended the analyzed region of parameter values. Still, the model has many parameters and the analysis is typically done by changing one or two parameters at a time. Thus, the work shows how HGT can indeed promote multistability, but it remains hard to grasp whether it consistently does so across a large region of the parameter values space.

---

## [Referee Report · Reviewer #3 (Public review)]

Hong et al. used a model they previously developed to study the impact of plasmid transfer on microbial multispecies communities. They investigated the effect of plasmid transfer on the existence of alternative stable states in a community. The model most closely resembles plasmid conjugation, where the transferred genes confer independent growth-related fitness effects and different plasmids do not affect each other's transfer or growth effects. For this process, the authors find that increasing the rate of plasmid transfer leads to an increasing number of stable states, as long as the model includes a constant death/dilution term.

This is an interesting and important topic, and I welcome the authors' efforts to explore these topics with mathematical modeling. The addition of sensitivity analyses also strengthens the usefulness for quantitative microbial ecologists. However, the additional sections have made the main text harder to read. Between the effect of the dilution rate, the increase in subpopulations with HGT, and the modulation of interspecies competition, the reviewers have suggested a number of factors that may explain the way plasmid transfer modulates multistability. I think it would be helpful if the authors could summarize some of these effects/interactions between different parameters in their model more. I personally continue to find the model very unintuitive, especially in the way it averages over subpopulations carrying more than one foreign plasmid. Additional sentences that give the reader intuition for the sensitivity analyses and how these interplay with the results would be good.

Specific points

(1) The model makes strong assumptions about the biology of HGT, that could be spelled out even more. Since the model is primarily applicable to HGT driven by the exchange of plasmids, I believe the abstract (and perhaps even the title of the paper) should be updated to reflect that.

(2) I am not surprised that a mechanism that creates diversity will lead to more alternative stable states. Specifically, the null model for the absence of HGT is to set gamma to zero, resulting in pij=0 for all subpopulations (line 454). This means that a model with N2 classes is effectively reduced to N classes. It seems intuitive that an LV-model with many more species would also allow for more alternative stable states. For a fair comparison one would really want to initialize these subpopulations in the model (with the same growth rates - e.g. mu1(1+lambda2)) but without gene mobility.

[Update:] It is good that it seems that initializing pij with non-zero abundance did not seem to affect the conclusion that higher amounts of HGT increases multi stability. However, rather than listing it as one control for a specific condition, I would argue that this is the appropriate null model across the board (where HGT rate is varied from 0 to a high value), including figures S9 and S10.

(3) The possibility that the same cell may be counted in different pij runs counter to all intuition that researchers coming from a background of compartmental /epidemiological modeling may have. The associated assumption that plasmids do not affect each other's dynamics or (growth/interaction) effects at all is also a very strong assumption. This should be signaled much earlier in the manuscript, possibly already in line 106 when the model is introduced.

---

## [Author Response]

The following is the authors’ response to the original reviews.

**Reviewer #1 (Public review):**
Summary:The authors present a modelling study to test the hypothesis that horizontal gene transfer (HGT) can modulate the outcome of interspecies competition in microbiomes, and in particular promote bistability in systems across scales. The premise is a model developed by the same authors in a previous paper where bistability happens because of a balance between growth rates and competition for a mutual resource pool (common carrying capacity). They show that introducing a transferrable element that gives a "growth rate bonus" expands the region of parameter space where bistability happens. The authors then investigate how often (in terms of parameter space) this bistability occurs across different scales of complexity, and finally under selection for the mobile element (framed as ABR selection).Strengths:The authors tackle an important, yet complex, question: how do different evolutionary processes impact the ecology of microbial ecosystems? They do a nice job at increasing the scales of heterogeneity and asking how these impact their main observable: bistability.

We appreciate the reviewer for agreeing with the potential value of our analysis. We are also grateful for the constructive comments and suggestions on further analyzing the influence of the model structure and the associated assumptions. We have fully addressed the raised issues in the updated manuscript and below.

Weaknesses:The author's starting point is their interaction LV model and the manuscript then explores how this model behaves under different scenarios. Because the structure of the model and the underlying assumptions essentially dictate these outcomes, I would expect to see much more focus on how these two aspects relate to the specific scenarios that are discussed. For example:A key assumption is that the mobile element conveys a multiplicative growth rate benefit (1+lambda). However, the competition between the species is modelled as a factor gamma that modulates the competition for overall resource and thus appears in the saturation term (1+ S1/Nm + gamma2*S2/Nm). This means that gamma changes the perceived abundance of the other species (if gamma > 1, then from the point of view of S1 it looks like there are more S2 than there really are). Most importantly, the relationship between these parameters dictates whether or not there will be bistability (as the authors state).This decoupling between the transferred benefit and the competition can have different consequences. One of them is that - from the point of view of the mobile element - the mobile element competes at different strengths within the same population compared to between. To what degree introducing such a mobile element modifies the baseline bistability expectation thus strongly depends on how it modifies gamma and lambda.Thus, this structural aspect needs to be much more carefully presented to help the reader follow how much of the results are just trivial given the model assumptions and which have more of an emergent flavour. From my point of view, this has an important impact on helping the reader understand how the model that the authors present can contribute to the understanding of the question "how microbes competing for a limited number of resources stably coexist". I do appreciate that this changes the focus of the manuscript from a presentation of simulation results to more of a discussion of mathematical modelling.

We thank the reviewer for the insightful suggestions. We agree with the reviewer that the model structure and the underlying assumptions need to be carefully discussed, in order to understand the generality of the theoretical predictions. In particular, the reviewer emphasized that how HGT affects bistability might depend on how mobile genetic elements modified growth rates and competition. In the main text, we have shown that when mobile genes only influence species growth rates, HGT is expected to promote multistability (Fig. 1 and 2). However, when mobile genes modify species interactions, the effect of HGT on multistability is dependent on how mobile genes change competition strength (Fig. 3a to f). When mobile genes increase competition, HGT promotes multistability (Fig. 3c and e). In contrast, when mobile genes relax competition, HGT is expected to reduce multistability (Fig. 3d and f).

In light of the reviewer’s comments, we have further generalized the model structure, by accounting for the scenario where mobile genes simultaneously modify growth rates and competition. The effect of mobile genes on growth rates is represented by the magnitude of 𝜆’s, and the influence on competition is described by another parameter 𝛿. By varying these two parameters, we can evaluate how the model structure and the underlying assumptions affect the baseline expectation. We performed additional simulations with broad ranges of 𝜆 and 𝛿 values. In particular, we analyzed whether HGT would promote the likelihood of bistability in two-species communities compared with the scenario without gene transfer (Fig. 3g-i). Our results suggested that: (1) With or without HGT, reducing 𝜆 (increasing neutrality) promotes bistability; (2) With HGT, increasing 𝛿 promotes bistability; (2) Compared with the population without HGT, gene transfer promotes bistability when 𝛿 is zero or positive, while reduces bistability when 𝛿 is largely negative. These results agree with the reviewer’s comment that the baseline bistability expectation depends on how HGT modifies gamma and lambda. In the updated manuscript, we have thoroughly discussed how the model structure and the underlying assumptions can influence the predictions (line 238-253).

We further expanded our analysis, by calculating how other parameters, including competition strength, growth rate ranges, and death/dilution rate, would affect the multistability of communities undergoing horizontal gene transfer (Fig. S2, S3, S9, S10, S11, S12, S13, S15). Together with the results presented in the first draft, these analysis enables a more comprehensive understanding of how different mechanisms, including but not limited to HGT, collectively shaped community multistability. In the updated manuscript, the reviewer can see the change of focus from exploring the effects of HGT to a more thorough discussion of the mathematical model. The revised texts highlighted in blue and the supplemented figures reflect such a change.

**Reviewer #2 (Public review):**
Summary:In this work, the authors use a theoretical model to study the potential impact of Horizontal Gene Transfer on the number of alternative stable states of microbial communities. For this, they use a modified version of the competitive Lotka Volterra model-which accounts for the effects of pairwise, competitive interactions on species growth-that incorporates terms for the effects of both an added death (dilution) rate acting on all species and the rates of horizontal transfer of mobile genetic elements-which can in turn affect species growth rates. The authors analyze the impact of horizontal gene transfer in different scenarios: bistability between pairs of species, multistability in communities, and a modular structure in the interaction matrix to simulate multiple niches. They also incorporate additional elements to the model, such as spatial structure to simulate metacommunities and modification of pairwise interactions by mobile genetic elements. In almost all these cases, the authors report an increase in either the number of alternative stable states or the parameter region (e.g. growth rate values) in which they occur.In my opinion, understanding the role of horizontal gene transfer in community multistability is avery important subject. This manuscript is a useful approach to the subject, but I'm afraid that a thorough analysis of the role of different parameters under different scenarios is missing in order to support the general claims of the authors. The authors have extended their analysis to increase their biological relevance, but I believe that the analysis still lacks comprehensiveness.Understanding the origin of alternative stable states in microbial communities and how often they may occur is an important challenge in microbial ecology and evolution. Shifts between these alternative stable states can drive transitions between e.g. a healthy microbiome and dysbiosis. A better understanding of how horizontal gene transfer can drive multistability could help predict alternative stable states in microbial communities, as well as inspire novel treatments to steer communities towards the most desired (e.g. healthy) stable states.Strengths:(1) Generality of the model: the work is based on a phenomenological model that has been extensively used to predict the dynamics of ecological communities in many different scenarios.(2) The question of how horizontal gene transfer can drive alternative stable states in microbial communities is important and there are very few studies addressing it.

We thank the reviewer for the positive comments on the potential novelty and conceptual importance of our work. We are also grateful for the constructive suggestions on the generality and comprehensiveness of our analysis. In particular, we agree with the reviewer that a thorough analysis of the role of different parameter could further improve the rigor of this work. We have fully addressed the raised issues in the updated manuscript and below.

Weaknesses:(1) There is a need for a more comprehensive analysis of the relative importance of the different model parameters in driving multistability. For example, there is no analysis of the effects of the added death rate in multistability. This parameter has been shown to determine whether a given pair of interacting species exhibits bistability or not (see e.g. Abreu et al 2019 Nature Communications 10:2120). Similarly, each scenario is analyzed for a unique value of species interspecies interaction strength-with the exception of the case for mobile genetic elements affecting interaction strength, which considers three specific values. Considering heterogeneous interaction strengths (e.g. sampling from a random distribution) could also lead to more realistic scenarios - the authors generally considered that all species pairs interact with the same strength. Analyzing a larger range of growth rates effects of mobile genetic elements would also help generalize the results. In order to achieve a more generic assessment of the impact of horizontal gene transfer in driving multistability, its role should be systematically compared to the effects of the rest of the parameters of the model.

We appreciate the suggestions. For each of the parameters that the reviewer mentioned, we have performed additional simulations to evaluate its importance in driving multistability.

For the added death rate, we have calculated the bistability feasibility of two-species populations under different values of 𝐷. Our results suggested that (1) varying death rate indeed changed the bistability probability of the system; (2) when the death rate was zero, mobile genetic elements that only modify growth rates would have no effects on system’s bistability. These results highlighted the importance of added death rate in driving multistability (Fig. S2, line 136-142).

For the interspecies interaction strength, we first extended our analysis on two-species populations. By calculating the bistability probability under different values of 𝛾, we showed that when interspecies interaction strength was smaller than 1, the influence of HGT on population bistability became weak (Fig. S3, line 143-147). We also considered heterogenous interaction strengths in multispecies communities, by randomly sampling 𝛾*ij* values from uniform distributions. While our results suggested the heterogeneous distribution of 𝛾*ij* didn’t fundamentally change the main conclusion, the mean value and variance of 𝛾*ij* affected the influence of HGT on multistability. The effects of HGT on community multistability becomes stronger when the mean value of 𝛾*ij* gets larger than 1 and the variance of 𝛾*ij* is small (Fig. S12, line 190-196).

We also analyzed different ranges of growth rates effects of mobile genetic elements. In particular, we sampled 𝜆*ij* values from uniform distributions with given widths. Greater width led to larger range of growth rate effects. We used five-species populations as an example and tested different ranges. Our results suggested that multistability was more feasible when the growth rate effects of MGEs were small. The qualitative relationship between HGT and community was not dependent on the range of growth rate effects (Fig. S13, line 197-205).

(2) The authors previously developed this theoretical model to study the impact of horizontal gene transfer on species coexistence. In this sense, it seems that the authors are exploring a different (stronger interspecies competition) range of parameter values of the same model, which could potentially limit novelty and generality.

We appreciate the comment. In a previous work (PMID: 38280843), we developed a theoretical model that incorporated horizontal gene transfer process into the classic LV framework. This model provides opportunities to investigate the role of HGT in different open questions of microbial ecology. In the previous work, we considered one fundamental question: how competing microbes coexist stably. In this work, however, we focused on a different problem: how alternative stable states emerge in complex communities. While the basic theoretical tool that we applied in the two works were similar, the scientific questions, application contexts and the implications of our analysis were largely different. The novelty of this work arose from the fact that it revealed the conceptual linkage between alternative stable states and a ubiquitous biological process, horizontal gene transfer. This linkage is largely unknown in previous studies. Exploring such a linkage naturally required us to consider stronger interspecies competitions, which in general would diminish coexistence but give rise to multistability. We believe that the analysis performed in this work provide novel and valuable insights for the field of microbial ecology.

With all the supplemented simulations that we carried out in light of the all the reviewer’s comments, we believe the updated manuscript also provide a unified framework to understand how different biological processes collectively shaped the multistability landscape of complex microbiota undergoing horizontal gene transfer. The comprehensive analyses performed and the diverse scenarios considered in this study also contribute to the novelty and generality of this work.

(3) The authors analyze several scenarios that, in my opinion, naturally follow from the results and parameter value choices in the first sections, making their analysis not very informative. For example, after showing that horizontal gene transfer can increase multistability both between pairs of species and in a community context, the way they model different niches does not bring significantly new results. Given that the authors showed previously in the manuscript that horizontal gene transfer can impact multistability in a community in which all species interact with each other, one might expect that it will also impact multistability in a larger community made of (sub)communities that are independent of (not interacting with) each-which is the proposed way for modelling niches. A similar argument can be made regarding the analysis of (spatially structured) metacommunities. It is known that, for smaller enough dispersal rates, space can promote regional diversity by enabling each local community to remain in a different stable state. Therefore, in conditions in which the impact of horizontal gene transfer drives multistability, it will also drive regional diversity in a metacommunity.

Thanks. Based on the reviewer’s comments, we have move Fig. 3 and 4 to Supplementary Information. In the updated manuscript, we have focused more on analyzing the roles of different parameters in shaping community multistability.

(4) In some cases, the authors consider that mobile genetic elements can lead to ~50% growth rate differences. In the presence of an added death rate, this can be a relatively strong advantage that makes the fastest grower easily take over their competitors. It would be important to discuss biologically relevant examples in which such growth advantages driven by mobile genetic elements could be expected, and how common such scenarios might be.

We appreciate the suggestion. Mobile genetic elements can drive large growth rate differences when they encode adaptative traits like antibiotic resistance (line 197-198).

We also analyzed different ranges of growth rates effects of mobile genetic elements, by sampling 𝜆*ij* values from uniform distributions with given widths. Our results suggested that multistability was more feasible when the fitness effects of MGEs were small (Fig. S13b). The qualitative relationship between HGT and community was not dependent on the range of growth rate effects (Fig. S13a and b). We discussed these results in line 197-205 of the updated main text.

**Reviewer #3 (Public review):**
Hong et al. used a model they previously developed to study the impact of horizontal gene transfer (HGT) on microbial multispecies communities. They investigated the effect of HGT on the existence of alternative stable states in a community. The model most closely resembles HGT through the conjugation of incompatible plasmids, where the transferred genes confer independent growth-related fitness effects. For this type of HGT, the authors find that increasing the rate of HGT leads to an increasing number of stable states. This effect of HGT persists when the model is extended to include multiple competitive niches (under a shared carrying capacity) or spatially distinct patches (that interact in a grid-like fashion). Instead, if the mobile gene is assumed to reduce between-species competition, increasing HGT leads to a smaller region of multistability and fewer stable states. Similarly, if the mobile gene is deleterious an increase in HGT reduces the parameter region that supports multistability.This is an interesting and important topic, and I welcome the authors' efforts to explore these topics with mathematical modeling. The manuscript is well written and the analyses seem appropriate and well-carried out. However, I believe the model is not as general as the authors imply and more discussion of the assumptions would be helpful (both to readers + to promote future theoretical work on this topic). Also, given the model, it is not clear that the conclusions hold quite so generally as the authors claim and for biologically relevant parameters. To address this, I would recommend adding sensitivity analyses to the manuscript.

We thank the reviewer for the agreeing that our work addressed an important topic and was wellconducted. We are also grateful for the suggestion on sensitivity analysis, which is very helpful to improve the rigor and generality of our conclusion. All the raised issues have been fully addressed in the updated manuscript and below.

Specific points(1) The model makes strong assumptions about the biology of HGT, that are not adequately spelled out in the main text or methods, and will not generally prove true in all biological systems. These include:a) The process of HGT can be described by mass action kinetics. This is a common assumption for plasmid conjugation, but for phage transduction and natural transformation, people use other models (e.g. with free phage that adsorp to all populations and transfer in bursts).b) A subpopulation will not acquire more than one mobile gene, subpopulations can not transfer multiple genes at a time, and populations do not lose their own mobilizable genes. [this may introduce bias, see below].c) The species internal inhibition is independent of the acquired MGE (i.e. for p1 the self-inhibition is by s1).These points are in addition to the assumptions explored in the supplementary materials, regarding epistasis, the independence of interspecies competition from the mobile genes, etc. I would appreciate it if the authors could be more explicit in the main text about the range of applicability of their model, and in the methods about the assumptions that are made.

We are grateful for the reviewer’s suggestions. In main text and methods of the updated manuscript, we have made clear the assumptions underlying our analysis. For point (a), we have clarified that our model primarily focused on plasmid transfer dynamics (line 74, 101, 517). Therefore, the process of HGT can be described by mass action kinetics, which is commonly assumed for plasmid transfer (line 537-538). For point (b), our model allows a cell to acquire more than one mobile genes. Please see our response to point (3) for details. We have also made it clear that we assumed the populations would not lose their own mobile gene completely (line 526-527). For (c), we have also clarified it in the updated manuscript (line 111-112, 527-528).

We have also performed a series of additional simulations to show the range of applicability of our model. In particular, we discuss the role of other mechanisms, including interspecies interaction strength, the growth rate effects of MGEs, MGE epistasis and microbial death rates in shaping the multistability of microbial communities undergoing HGT. These results were provided in Fig. S2, S3, S9, S10, S11, S12, S13 and S15.

(2) I am not surprised that a mechanism that creates diversity will lead to more alternative stable states. Specifically, the null model for the absence of HGT is to set gamma to zero, resulting in pij=0 for all subpopulations (line 454). This means that a model with N^2 classes is effectively reduced to N classes. It seems intuitive that an LV-model with many more species would also allow for more alternative stable states. For a fair comparison, one would really want to initialize these subpopulations in the model (with the same growth rates - e.g. mu1(1+lambda2)) but without gene mobility.

We appreciate the insightful comments. The reviewer was right that in our model HGT created additional subpopulations in the community. However, with or without HGT, we calculated the species diversity and multistability based on the abundances of the 𝑁 species (*si* in our model), instead of all the *pij* subpopulations. Therefore, although there exist more ‘classes’ in the model with HGT, the number of ‘classes’ considered when we calculated community diversity and multistability was equal. In light of the reviewer’s suggestion, we have also performed additional simulations, where we initialized the subpopulations in the model with nonzero abundances. Our results suggested that initializing the *pij* subpopulations with non-zero abundances didn’t change the main conclusion (Fig. S11, line 188-189).

(3) I am worried that the absence of double gene acquisitions from the model may unintentionally promote bistability. This assumption is equivalent to an implicit assumption of incompatibility between the genes transferred from different species. A highly abundant species with high HGT rates could fill up the "MGE niche" in a species before any other species have reached appreciable size. This would lead to greater importance of initial conditions and could thus lead to increased multistability.This concern also feels reminiscent of the "coexistence for free" literature (first described here http://dx.doi.org/10.1016/j.epidem.2008.07.001) which was recently discussed in the context of plasmid conjugation models in the supplementary material (section 3) of https://doi.org/10.1098/rstb.2020.0478 .

We appreciate the comments. Our model didn’t assume the incompatibility between MGEs transferred from different species. Instead, it allows a cell to acquire more than one MGEs. In our model, *pij* described the subpopulation in the 𝑖-th species that acquired the MGE from the 𝑗th species. Here, *pij* can have overlaps with *pik* (𝑗 ≠ 𝑘). In other words, a cell can belong to *pij* and *pik* at the same time. The *pij* subpopulation is allowed to carry the MGEs from the other species. In the model, we used ∏k≠i,j(1+λikpiksi) to describe the influence of the other MGEs on the growth of *pij*.

We also thank the reviewer for bringing two papers into our attention. We have cited and discussed these papers in the updated manuscript (line 355-362).

(4) The parameter values tested seem to focus on very large effects, which are unlikely to occur commonly in nature. If I understand the parameters in Figure 1b correctly for instance, lambda2 leads to a 60% increase in growth rate. Such huge effects of mobile genes (here also assumed independent from genetic background) seem unlikely except for rare cases. To make this figure easier to interpret and relate to real-world systems, it could be worthwhile to plot the axes in terms of the assumed cost/benefit of the mobile genes of each species.

Thanks for the comments. In the main text, we presented one simulation results that assumed relatively large effects of MGE on species fitness, as the reviewer pointed out. In the updated manuscript, we have supplemented numerical simulations that considered different ranges of fitness effects, including the fitness effect as small as 10% (Fig. S13a). We have also plotted the relationship between community multistability and the assumed fitness effects of MGEs, as the reviewer suggested (Fig. S13b). Our results suggested that multistability was more feasible when the fitness effects of MGEs were small, and changing the range of MGE fitness effects didn’t fundamentally change our main conclusion. These results were discussed in line 197-205 of the updated main text.

Something similar holds for the HGT rate (eta): given that the population of *E. coli* or Klebsiella in the gut is probably closer to 10^9 than 10^12 (they make up only a fraction of all cells in the gut), the assumed rates for eta are definitely at the high end of measured plasmid transfer rates (e.g. F plasmid transfers at a rate of 10^-9 mL/CFU h-1, but it is derepressed and considered among the fastest - https://doi.org/10.1016/j.plasmid.2020.102489). To adequately assess the impact of the HGT rate on microbial community stability it would need to be scanned on a log (rather than a linear) scale. Considering the meta-analysis by Sheppard et al. it would make sense to scan it from 10^-7 to 1 for a community with a carrying capacity around 10^9.

We thank the reviewer for the constructive suggestion. We have carried out additional simulations by scanning the 𝜂 value from 10^-7^ to 1. The results suggested that increasing HGT rates started to promote multistability when 𝜂 value exceeded 10^-2^ per hour (Fig. S9, line 337-346). This corresponds to a conjugation efficiency of 10^-11^
*cell*^-1^ ∙ *mL*^-1^∙ *mL* when the maximum carrying capacity equals 10^9^
*cells* ∙ *mL*^-1^, or a conjugation efficiency of 10^-14^
*cell*^-1^ ∙ *hr*^-1^∙ *mL* when the maximum carrying capacity equals 10^12^
*cells* ∙ *mL*^-1^.

(5) It is not clear how sensitive the results (e.g. Figure 2a on the effect of HGT) are to the assumption of the fitness effect distribution of the mobile genes. This is related to the previous point that these fitness effects seem quite large. I think some sensitivity analysis of the results to the other parameters of the simulation (also the assumed interspecies competition varies from figure to figure) would be helpful to put the results into perspective and relate them to real biological systems.

We appreciate the comments. In light of the reviewer’s suggestion, we have changed the range of the fitness effects and analyzed the sensitivity of our predictions to this range. As shown in Fig. S13, changing the range of MGE fitness effects didn’t alter the qualitative interplay between HGT and community multistability. We have also examined the sensitivity of the results to the strength of interspecies competition strength (Fig. S3, S10, S12). These results suggested that while the strength of interspecies interactions played an important role in shaping community multistability, the relationship between HGT rate and multistability was not fundamentally changed by varying interaction strength. In addition, we examined the role of death rates (Fig. S2). In the updated manuscript, we discussed the sensitivity of our prediction to these parameters in line 136-147, 190205, 335-354.

**Recommendations for the authors:**

**Reviewer #2 (Recommendations for the authors):**
Please find below a few suggestions that, in my opinion, could help improve the manuscript.TITLEIt might not be clear what I 'gene exchange communities' are. Perhaps it could be rewritten for more specificity (e.g. '...communities undergoing horizontal gene transfer').

We have updated the title as the reviewer suggested.

ABSTRACTThe abstract could also be edited to improve clarity and specificity. Terms like 'complicating factors' are vague, and enumerating specific factors would be better. The results are largely based on simulations, no analytical results are plotted, so I find that the sentence starting with 'Combining theoretical derivation and numerical simulations' can be a bit misleading.

We appreciate the suggestions. We have enumerated the specific factors and scenarios in the updated abstract (line 18-26). We have also replaced 'Combining theoretical derivation and numerical simulations' with ‘Combining mathematical modeling and numerical simulations’.

INTRODUCTION- Line 42, please revise this paragraph. The logical flow is not so clear, it seems a bit like a list of facts, but the main message might not be clear enough. Also, it would be good to define 'hidden' states or just rewrite this sentence.

We appreciate the suggestion. In the updated manuscript, we have rewritten this paragraph to improve the logical flow and clarity (line 46-52).

- Line 54, there is little detail about both theoretical models and HGT in this paragraph, and mixing the two makes the paragraph less focused. I suggest to divide into two paragraphs and expand its content. For example, you could explain a bit some relevant implications of MGE.

We appreciate the suggestion. In the updated manuscript, we have divided this paragraph into two paragraphs, focusing on theoretical models and HGT, respectively (line 55-71). In particular, we have added explanations on the implications of MGEs (line 66-69), as the reviewer suggested.

- Line 72, as mentioned in the abstract, it would be better to explicitly mention which confounding factors are going to be discussed.

Thanks for the suggestion. We have rewritten this part as “We further extended our analysis to scenarios where HGT changed interspecies interactions, where microbial communities were subjected to strong environmental selections and where microbes lived in metacommunities consisting of multiple local habitats. We also analyzed the role of different mechanisms, including interspecies interaction strength, the growth rate effects of MGEs, MGE epistasis and microbial death rates in shaping the multistability of microbial communities. These results created a comprehensive framework to understand how different dynamic processes, including but not limited to HGT rates, collectively shaped community multistability and diversity” (line 75-82).

RESULTS- The basic concepts (line 77) should be explained with more detail, keeping the non-familiar reader in mind. The reader might not be familiar with the concept of bistability in terms of species abundance. Also, note that mutual inhibition does not necessarily lead to positive feedback, as an interaction strength between 0 and 1 might still be considered inhibition. In any case, in Figure 1 it is not obvious how the positive feedback is represented, the caption should explain it. Note that neither the main text nor the caption explains the metaphor of the landscape and the marble that you are using in Figure 1a.

We have rewritten this paragraph to provide more details on the basic concepts (line 86-99). We have removed the statement about ‘mutual inhibition’ to avoid being misleading. We have also updated the caption of Fig. 1a to explain the metaphor of the landscape and the marble (line 389396).

- In the classical LV model, bistability does not depend on growth rates, but only on interaction strength. Therefore, I think that much of the results are significantly influenced by the added death rate. I believe that if the death rate is set to zero, mobile genetic elements that only modify growth rates will have no effect on the system's bistability. Because of this, I think that a thorough analysis of the role of the added death (dilution) rate and the distribution of growth rates is especially needed.

We are grateful for the reviewer’s insightful comments. In the updated manuscript, we have thoroughly analyzed the role of the added death (dilution) rate on the bistability of communities composed of two species (Fig. S2). Indeed, as the reviewer pointed out, if the death rate equals zero, mobile genetic elements that only modify growth rates will have no effect on the system's bistability. We have discussed the role of death rate in line 136-142 of the updated manuscript.

We have also expanded our analysis on the distribution of growth rates. In particular, we considered different ranges of growth rates effects of mobile genetic elements, by sampling 𝜆_*ij*_ values from uniform distributions with given widths (Fig. S13). Greater width led to larger range of growth rate effects. We used five-species populations as an example and tested different ranges.

Our results suggested that multistability was more feasible when the growth rate effects of MGEs were small (Fig. S13b). The qualitative relationship between HGT and community was not dependent on the range of growth rate effects (Fig. S13a). These results are discussed in line 197205 of the updated manuscript.

- The analysis uses gamma values that, in the absence of an added death rate, render a species pair bistable. Therefore, multistability would be quite expected for a 5 species community. Note that, multistability is possible in communities of more than 2 species even if all gamma values are smaller than 1. Analyzing a wide range of interaction strength distributions would really inform on the relative role of HGT in multistability across different community scenarios.

We are grateful for the reviewer’s suggestion. In light of the reviewer’s comments, in the updated manuscript, we have performed additional analysis by focusing on a broader range of interaction strengths (Fig. S3, S10, S12), especially the gamma values below 1 (Fig. S10). Our results agreed with the reviewer’s notion that multistability was possible in communities of more than 2 species even if all gamma values were smaller than 1 (Fig. S10).

- I would recommend the authors extend the analysis of the model used for Figures 1 and 2. Figures 3 and 4 could be moved to the supplement (see my point in the public review), unless the authors extend the analysis to explain some non-intuitive outcomes for niches and metacommunities.

Thanks. In the updated manuscript we have performed additional simulations to extend the analysis in Figure 1 and 2. These results were presented in Fig. S2, S3, S9, S10, S11, S12, and S13. We have also moved Figure 3 and 4 to SI as the reviewer suggested.

- The authors seem to refer to fitness and growth rates as the same thing. This could lead to confusion - the strongest competitor in a species pair could also be interpreted as the fittest species despite being the slowest grower. I think there's no need to use fitness if they refer to growth rates. In any case, they should define fitness if they want to use this concept in the text.

We are grateful for the insightful suggestion. To avoid confusion, we have used ‘growth rate’ throughout the updated manuscript.

- Across the text, the language needs some revision for clarity, specificity, and scientific style. In lines 105 - 109 there are some examples, like the use of 'in a lot of systems', and ' interspecies competitions' (I believe they mean interspecies interaction strengths).

We appreciate the reviewer for pointing them out. We have thoroughly checked the text and made the revisions whenever applicable to improve the clarity and specificity.

- Many plots present the HGT rate on the horizontal axis. Could the authors explain why is it that the rate of HGT is relatively important for the number of alternative stable states? I understand how from zero to a small positive number there is a qualitative change. Beyond that, it shouldn't affect bistability too much, I think. If I am right, then other parameters could be more informative to plot in the horizontal axis. If I am wrong, I think that providing an explanation for this would be valuable.

Thanks. To address the reviewer’s comment, we have systematically analyzed the effects of HGT on community multistability, by scanning the HGT rate from 10^-7^ to 10^0^*hr*^-1^ . In communities of two or multiple species, our simulation results showed that multistability gradually increased with HGT rate when HGT rate exceeded 10^2^*hr*^-1^. These results, presented in Fig. S9 and discussed in line 337-346, provided a more quantitative relationship between multistability and HGT rate.

While in this work we showed the potential role of HGT in modulating community multistability, our results didn’t exclude the role of the other parameters. Motivated by the comments raised by the reviewers, in the updated manuscript, we have performed additional simulations to analyze the influence of other parameters in shaping community multistability. These parameters include death or dilution rate (Fig. S2), interaction strength (Fig. S3, S9, S10, S11, S12, S14, S15), 𝜆 range (Fig. S13, S15) and 𝛿 value (Fig. 3g, h, i). In many of the supplemented results (Fig. S2b, S3b, S13b, Fig. 3g, 3h and 3i), we have also plotted the data by using these parameters as the x axis. We believe the updated work now provided a more comprehensive framework to understand how different mechanisms, including but not limited to HGT, might shape the multistability of complex microbiota. These points were discussed in line 136-147, 190-205, 238-253, 334-354 of the updated main text.

- My overall thoughts on the case of antibiotic exposure are similar to those of previous sections. Very few of the different parameters of the model are analyzed and discussed. In this case, the authors increased the interaction strength to ~0.4 times higher compared to previous sections. Was this necessary, and why?

Thanks for the comments. In the previous draft, the interaction strength 𝛾=1.5 was tested as an example. Motivated by the reviewer’s comments, in the updated manuscript, we have examined different interaction strengths, including the strength (𝛾 = 1.1) commonly tested in other scenarios. The prediction equally held for different 𝛾 values (Fig. S15). We have also analyzed different 𝜆 ranges (Fig. S15). These results, together with the analyses presented in the earlier version of the manuscript, suggested the potential role of HGT in promoting multistability for communities under strong selection. The supplemented results were presented in Fig. S15 and discussed in line 293-295 of the updated manuscript.

- Line 195, if a gene encodes for the production of a public good, why would its HGT reduce interaction strength? I can think of the opposite scenario: the gene is a public good, and without HGT there is only one species that can produce it. Let's imagine that the public good is an enzyme that deactivates an antibiotic that is present in the environment, and then the species that produces has a positive interaction with another species in a pairwise coculture. If HGT happens, the second species becomes a producer and does not need the other one to survive in the presence of antibiotics anymore. The interaction can then become more competitive, as e.g. competition for resources could become the dominant interaction.

We are grateful for pointing it out. In the updated manuscript, we have removed this statement.

DISCUSSION- L 267 "by comparison with empirical estimates of plasmid conjugation rates from a previous study [42], the HGT rates in our analysis are biologically relevant in a variety of natural environments". The authors are using a normalized model and the relevance of other parameter values is not discussed. If the authors want to claim that they are using biologically relevant HGT, they should also discuss whether the rest of the parameter values are biologically relevant. I recommend relaxing this statement about HGT rates.

We appreciate the suggestion. We agree with the reviewer that other parameters including the death/dilution rate, interactions strength and 𝜆 ranges are also important in shaping community multistability. We have performed additional analysis to show the effects of these parameters. In light of the reviewer’s suggestion, we have relaxed this statement and thoroughly discussed the context-dependent effect of HGT as well as the roles of different parameters (line 334-354).

- Last sentence: "Therefore, inhibiting the MGE spread using small molecules might offer new opportunities to reshape the stability landscape and narrow down the attraction domains of the disease states". It is not clear what procedure/technique the authors are suggesting. If they want to keep this statement, the authors should give more details on how small molecules can be/are used to inhibit MGE.

We appreciated the comments. Previous studies have shown some small molecules like unsaturated fatty acids can inhibit the conjugative transfer of plasmids. By binding the type IV secretion traffic ATPase TrwD, these compounds limit the pilus biogenesis and DNA translocation. We have provided more details regarding this statement in the updated manuscripts (line 376-379).

METHODS- Line 439, mu_i should be presented as the maximum 'per capita' growth rate.

We have updated the definition of 𝜇*i* following the suggestion (line 529).

- Line 444, this explanation is hard to follow, please expand it to provide more details. You could provide an example, like explaining that all individuals from S1 have the MGE1 and therefore they have mu_1 = mu_01 ... After HGT, their fitness changes if they get the plasmid from S2, so a term lambda2 appears.

Thanks. In the updated manuscript, we have expanded the explanation by providing an example as the reviewer suggested (line 534-537).

- The normalization assumes a common carrying capacity Nm (Eqs 1-4) and then it's normalized (Eqs. 5-8). It would be better to start from a more general scenario in which each species has a different carrying capacity and then proceed with the normalization.

We appreciate the suggestion. In the updated manuscript, we have started our derivation from the scenario where each species has a different carrying capacity before proceeding with the normalization (section 1 of Methods, line 516-554). The same equations can be obtained after normalization.

- I think that the meaning of kappa (the plasmid loss rate) is not explained in the text.

Thanks for pointing it out. We have explained the meaning of kappa in the updated text (line 108, 154, 539-541, 586-587, 607).

SUPPLEMENT- Figure S4, what are the different colors in panel b?

In panel b of Fig. S4, the different colors represent the simulation results repeated with randomized growth rates. We have made it clear in the updated SI.

**Reviewer #3 (Recommendations for the authors):**
(1) Please extend your description of the model, so it is easier to understand for readers who have not read the first paper. Especially the choice to describe the model as species and subpopulations, as opposed to writing it as MGE-carrying and MGE-free populations of each species makes it quite complicated to understand which parameters influence each other.

Thanks for the suggestion. We have extended the model description in the updated manuscript, which provides a more detailed introduction on model configurations and parameter definitions (line 86-99, 101-113, 151-159). We have also updated the Methods to extend the model description.

(2) Please define gamma_ji in equation 13 and eta_jki in equation 14 (how to map the indices onto the assumed directionality of the interaction).

We have defined these two parameters in the updated manuscript (line 584-586, 630-632).

(3) Line 511: please add at the beginning of this paragraph that you are assuming a grid-like arrangement of patches which will be captured by dispersal term H.

We have updated this paragraph to make this assumption clear (line 636-637).

(4) Line 540: "used in our model" (missing a word).

We have corrected it in the updated manuscript.

(5) Currently the analyses looking at the types of growth effects HGT brings (Figures 5-7) feel very "tacked on". These are not just "confounding factors", but rather scenarios that are much more biologically realistic than the assumption of independent effects. I would introduce them earlier in the text, as I think many readers may not trust your results until they know this was considered (+ how it changes the conclusions).

We are grateful for the suggestion. We agree with the reviewer that these biologically realistic scenarios should be introduced earlier in the text. In the updated manuscript, we have moved these analyses forward, as sections 3, 4 and 5. We have also avoided the term “confounding factors”. Instead, in the updated manuscript, we have separated these analyses into different sections, and clearly described each scenario in the section title (line 217-218, 254, 275).

(6) In some places the manuscript refers to HGT, in others to MGE presence (e.g. caption of Figure 6). These are not generally the same thing, as HGT could also occur due to extracellular vesicles or natural transformation etc. Please standardize the nomenclature and make it clearer which type of processes the model describes.

We appreciate the comment. The model in this work primarily focused on the process of plasmid transfer. We have made it clear throughout the main text.

(7) In many figures the y-axis starts at a value other than 0. This is a bit misleading. In addition, I would recommend changing the title "Area of bistability region" to "Area of bistability" or perhaps even "Area of multistability" (since more than two species are considered).

Thanks for the suggestion. We have updated all the relevant figures to make sure that their y-axes start at 0. We have also changed the title “Area of bistability region” to “Area of multistability”, whenever it is applicable.

(8) Figure 7: what are the assumed fitness effects of the mobile genes in the simulation? Which distribution were they drawn from? Please add this info to the figure caption here and elsewhere.

In Figure 7, we explored an extreme scenario of the fitness effects of the mobile genes, where the population was subjected to strong environmental selection and only cells carrying the mobile gene could grow. Therefore, the carriage of the mobile gene changed the species growth rate from 0 to a positive value µ_i_. When calculating the number of stable states in the communities, we randomly drew the µ_i_ values from a uniform distribution between 0.3 and 0.7 *hr*^-1^. We had added this information in the figure caption (line 505-508) and method (line 615-617) of the updated manuscript.